# The microRNA miR-71 suppresses maladaptive UPR^mt signaling through both cell-autonomous and cell-non-autonomous mechanisms

Ina Kirmes[1], Grace Ching Ching Hung[1], Anne Hahn[1], Chuan-Yang Dai[1], Daniel Campbell [1], Arnaud Ahier [1], Rachel Shin Yie Lee[1], Alexander Palmer [1] & Steven Zuryn [1,2] ✉

Mitochondria play a central role in metabolism and biosynthesis, but function also as platforms that perceive and communicate environmental and physiological stressors to the nucleus and distal tissues. Systemic mitochondrial signaling is thought to synchronize and amplify stress responses throughout the whole body, but during severe or chronic damage, overactivation of mitochondrial stress pathways may be maladaptive and exacerbate aging and metabolic disorders. Here we uncover a protective micro(mi)RNA response to mtDNA damage in *Caenorhabditis elegans* that prolongs tissue health and function by interfering with mitochondrial stress signaling. Acting within muscle cells, we show that the miRNA miR-71 is induced during severe mitochondrial damage by the combined activities of DAF-16, HIF-1, and ATFS-1, where it restores sarcomere structure and animal locomotion by directly suppressing the inordinate activation of DVE-1, a key regulator of the mitochondrial unfolded protein response (UPR^mt). Indirectly, miR-71 also reduces the levels of multiple neuro- and insulin-like peptides and their secretion machinery, resulting in decreased cell-non-autonomous signaling of mitochondrial stress from muscle to glia cells. miR-71 therefore beneficially coordinates the suppression of both local and systemic mitochondrial stress pathways during severe organelle dysfunction. These findings open the possibility that metabolic disorders could be ameliorated by limiting the overactivation of mitochondrial stress responses through targeted small RNAs.

Mitochondria are compartmentalized hubs of cellular metabolism, biosynthesis, and signaling, and their dysregulation is closely associated with metabolic disorders and diseases of aging, including diabetes, neurodegeneration, and cancer[1,2]. Organisms frequently encounter both extrinsic and intrinsic stresses that perturb mitochondria and disrupt cellular homeostasis. An inevitable threat to correct mitochondrial function are mitochondrial genome (mtDNA) mutations and other molecular lesions, which accumulate in post-

[1]Clem Jones Centre for Ageing Dementia Research, Queensland Brain Institute, Faculty of Health, Medicine and Behavioural Sciences, The University of Queensland, Brisbane 4072, Australia. [2]NHMRC Centre for Research Excellence in Mechanisms in NeuroDegeneration - Alzheimer's Disease (MIND-AD CRE), The University of Queensland, Brisbane, Australia. ✉e-mail: s.zuryn@uq.edu.au

mitotic tissues through oxidative damage and duplication mistakes and can expand clonally in stem cell compartments[3,4]. mtDNA mutations can also be maternally inherited, culminating in congenital metabolic disorders. Both inherited and acquired mtDNA mutations can persist for the duration of an organism's life, causing varying amounts of chronic mitochondrial stress that may contribute to disease[5,6]. As such, the capacity of cells to perceive, respond proportionately, and function under mtDNA stress underpins organismal health, especially during aging and mitochondrial disease conditions.

Metazoans have evolved mitochondrial-nuclear signaling pathways, such as the mitochondrial unfolded protein response (UPR$^{mt}$)[7–11], that can alleviate mitochondrial burden and promote recovery and overall organismal health. Genes that are upregulated as part of the UPR$^{mt}$, which is activated by mtDNA lesions in both mammals and *C. elegans*[12–16], encode protein functions that encompass a broad profile of processes that extend beyond direct mitochondrial activity, including promoting innate immunity and pathogen clearance, maintaining cytosolic proteostasis, activating anerobic metabolism and reactive oxygen species detoxification, and mediating signaling between tissues[17–19]. Although mitochondria-nuclear communication mechanisms differ between mammals and invertebrates, they are regulated by conceptually similar processes[20]. In *C. elegans*, where the regulatory mechanisms of the UPR$^{mt}$ are best characterized, multiple factors mediate UPR$^{mt}$ signaling. Nuclear localization of the homeobox protein DVE-1 mediates a branch of the transcriptional response to mitochondrial stress[10] and its activity occurs downstream of chromatin remodeling driven by mitochondrial dysfunction. Upon mitochondrial stress, the histone methyltransferase MET-2 is required for nuclear translocation of the cytosolic protein LIN-65, a histone H3K9 methyltransferase that globally compacts chromatin while promoting the redistribution and transcriptional activity of DVE-1 in chromatin sites that remain open[21]. This suggests that DVE-1 may play a key role in the propagation of an epigenetic memory of mitochondrial stress experienced during early development that underpins changes in later life traits, such as adult lifespan[21,22].

In addition to retrograde signaling, mitochondrial stress can be communicated systemically in multiple species. In *C. elegans*, mitochondrial damage restricted to neurons can activate UPR$^{mt}$ responses in peripheral tissues that are not experiencing stress themselves, such as the intestine or germline[23,24], via cell-non-autonomous signaling mechanisms that involve the secretion of neurotransmitters such as glutamate, acetylcholine, tyramine, serotonin, as well as insulin-like peptides, neuropeptides, and Wnt/EGL-20[25–28]. This intercellular or inter-tissue communication of mitochondrial stress has been shown to occur in mammals and involves multiple circulating factors such as fibroblast growth factor 21 (FGF21), adrenomedullin 2 (ADM2), growth/differentiation factor 15 (GDF15), angiopoietin-like 6 (ANGPTL6), and various mitochondrial-derived peptides[29]. Intercellular communication of mitochondrial stress is thought to play a role in synchronizing and amplifying stress responses throughout the whole body, thereby coordinating homeostasis of the organism more effectively than if stress responses were restricted to within already affected cells. However, under chronic mitochondrial stress, localized as well as systemic activation of mitochondrial stress responses could be detrimental to organismal fitness. Indeed, whereas acute UPR$^{mt}$ signaling is important for mitochondrial recovery following stress, chronic UPR$^{mt}$ activation is maladaptive as it can drive the propagation of mutant mitochondrial genomes[14], induce or aggravate neuroinflammation and neurodegenerative disorders[30,31], and contribute to the pathogenesis of diseases of aging, including cancers[32]. For instance, subpopulations of cancer cells that demonstrate persistent activation of the UPR$^{mt}$ display an adaptive metastatic advantage, with breast cancer patients with high UPR$^{mt}$ genetic signatures having significantly worse clinical outcomes[32]. Mechanisms that counteract both cell-autonomous and cell-non-autonomous mitochondrial stress signaling during chronic mitochondrial dysfunction are yet to be understood but could play an essential role in organismal homeostasis and be important for ameliorating a broad spectrum of disorders.

Here we reveal a miRNA-mediated protective response to chronic mtDNA damage within the muscle of *C. elegans*, which acts to simultaneously attenuate cell-autonomous as well as cell-non-autonomous UPR$^{mt}$ signaling by targeting DVE-1 and systemic mitochondrial stress signaling mechanisms. We find that during mitochondrial stress, miR-71 is strongly upregulated through a combination of HIF-1, ATFS-1, and DAF-16 nuclear localization, where it then recognizes and targets *dve-1* transcripts for degradation. Overexpression of miR-71 restores muscle cell structure and function disrupted by chronic mtDNA damage and hyperactivated UPR$^{mt}$. Remarkably, severe mitochondrial stress suppresses a broad range of cell-cell signaling molecules in a miR-71-dependent manner, suggesting that miR-71 silences systemic signaling during chronic mitochondrial stress possibly to prevent maladaptive inter-tissue responses. Indeed, muscle miR-71 limits distal DVE-1 nuclear accumulation in glial cells, upstream mediators of whole animal stress signaling, via the neuropeptide-like protein *nlp-52*. Together, our results reveal a small non-coding RNA regulatory network that antagonizes local and systemic mitochondrial quality control programs during chronic stress.

## Results

### miR-35 family and miR-71 are induced during mitochondrial stress

The broad abilities of small non-coding RNAs to regulate gene expression in a manner that promotes homeostasis during stress prompted us to investigate whether a miRNA response to mtDNA damage is activated in animals. miRNAs are a class of 21- to 23-nucleotide non-coding RNAs that post-transcriptionally downregulate genes by targeting mRNAs for translational inhibition or degradation[33]. We performed small RNA sequencing in genetically engineered *C. elegans* backgrounds that incur muscle-specific mitochondrial DNA double strand breaks (mtDSBs) via the expression of a mitochondria-targeted endonuclease ($^{MTS}$*Pst*I) under the *myo-3* promoter (from hereon called *myo-3*p::mtDSB) (Fig. 1a, b)[13]. When compared to control animals expressing a catalytically inactive version of $^{MTS}$*Pst*I (*myo-3*p::mtDSB*), we identified 24 miRNAs that were significantly upregulated as a result of mtDNA lesion (FDR < 0.05, log2 fold change > 2, Supplementary Data Table 1). Interestingly, most of the members of the miR-35 family of miRNAs (miR-35/36/37/38/39/40/41-3p) were highly represented in this group (Fig. 1c). This miRNA family is widely conserved and its members share identical seed regions, and are therefore likely to target the same mRNAs and have similar biological functions[34,35], which are as diverse as embryonic patterning[36–38], sex determination[39], germ cell apoptosis[40], and hypoxia and starvation responses[41,42]. In addition to the miR-35 family, miR-71 was upregulated in response to mtDSBs (Fig. 1c). miR-71 has previously been shown to increase in abundance with age in *C. elegans*, and is indispensable for lifespan enhancements caused by starvation[43–45] or germline removal[46]. It has also recently been demonstrated to mediate olfactory regulation of organismal proteostasis and longevity[47]. We confirmed that miR-35 family members and miR-71 were upregulated in animals during mtDSBs using quantitative (q)PCR (Fig. 1d, Extended Data Fig. 1a).

In addition to mtDSBs, which can be caused by replication fork arrest during mtDNA synthesis and various endogenous and exogenous genotoxins[48], mtDNA mutations also pose a major risk to correct mitochondrial function when present at high enough heteroplasmy levels (the percentage of mitochondrial genomes carrying a mutation)[49]. We found that a large 3.1-kb mtDNA deletion (called *uaDf5*) that removes four protein-coding and seven tRNA genes (Fig. 1b)[50,51] caused upregulation of both miR-71 and miR-35 at heteroplasmy levels of 60% (Fig. 1e, Extended Data Fig. 1b). Animals

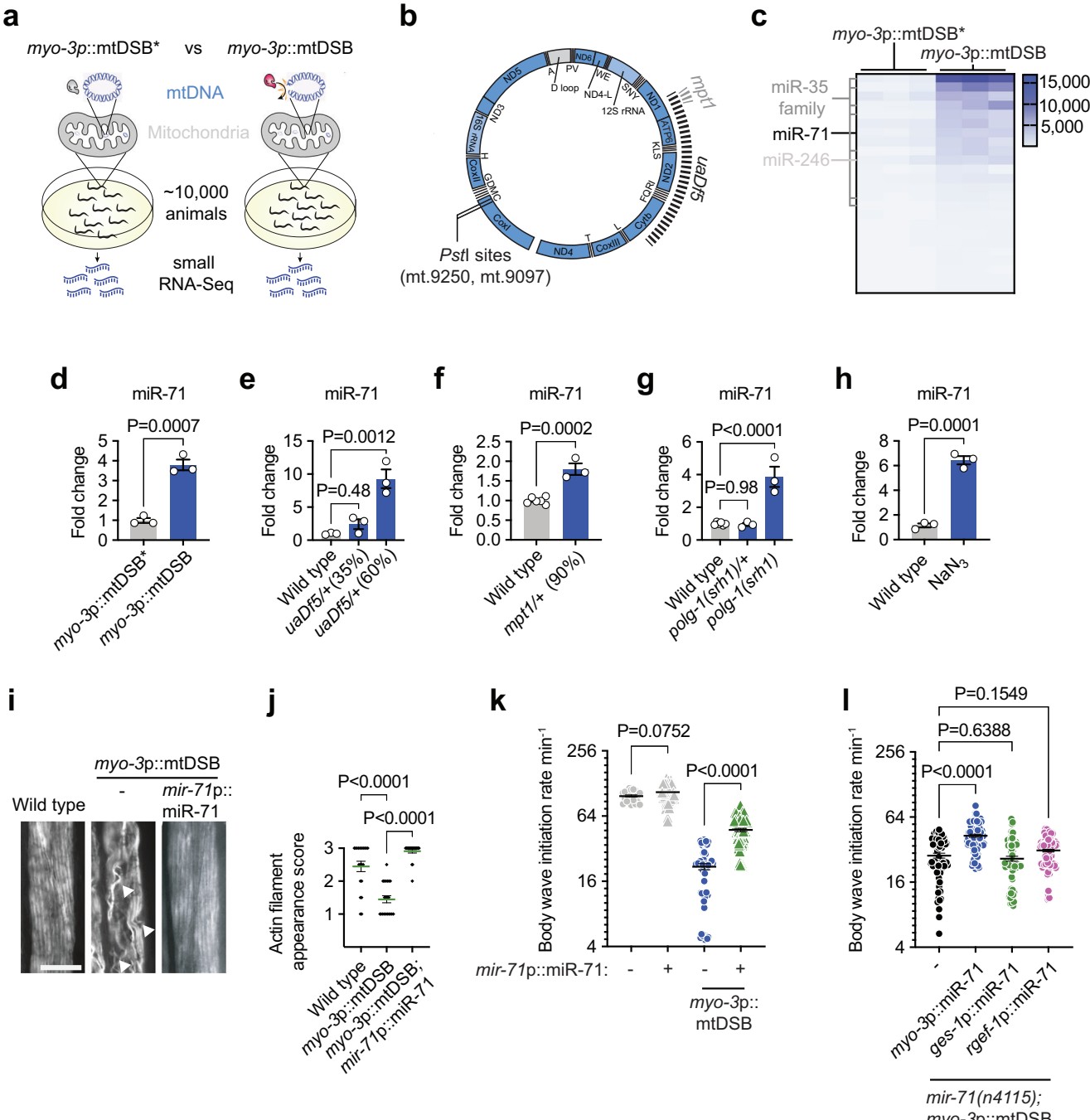

**Fig. 1 | miR-71 is induced by mitochondrial stress and protects muscle cells against mtDNA damage. a** populations of animals expressing *myo-3*p::mtDSB (the endonuclease *Pst*I fused to a mitochondrial targeting signal and expressed under a body wall muscle-specific promoter) and catalytically inactive *myo-3*p::mtDSB* were raised to the L4 stage and underwent small RNA sequencing. **b** map of *C. elegans* mtDNA showing *uaDf5* and *mpt1* deletions and the sites of *Pst*I cleavage. **c** heatmap of *C. elegans* miRNAs upregulated by mtDSBs incurred selectively in body wall muscle cells expressing mitochondrial-targeted *Pst*I (*myo-3*p::mtDSB). Comparisons were made between animals expressing *myo-3*p::mtDSB and *myo-3*p::mtDSB*. *n* = 3. **d–h** quantitative (q)PCR of miR-71 levels in animals that have (**d**) mtDSBs induced in muscle cells, *n* = 3, (**e**) the *uaDf5*/+ mtDNA mutation at either 35% or 60% heteroplasmy, *n* = 3, (**f**) the *mpt1*/+ mtDNA mutation at 90% heteroplasmy, *n* = 6 and 3, (**g**) a *polg-1(srh1)* mutation (heterozygous or homozygous),

*n* = 6, 3 and 3, and (**h**) been treated with sodium azide (NaN₃), *n* = 3. For (**d–h**) columns represent mean ± SEM; *n* ≥ 3; two-way Student's t test or one-way ANOVA with Tukey's post hoc test for experiments with either two or more than two comparisons, respectively. **i** representative photomicrographs of body wall muscle tissue in L4 animals stained with phalloidin. Arrow heads indicate disruptions to actin filament structure in animals expressing *myo-3*p::mtDSB. Scale bar, 10 μm. **j** quantification of actin appearance (see methods). Bars represent mean ± SEM; *n* = 22, 21 and 20; one-way ANOVA with Tukey's post hoc test. **k, l** quantification of body wave initiation rate (using WormLab automated software analysis) of L4 animals placed in liquid. For k, *n* = 21, 105, 56 and 64; l, *n* = 67, 95, 78 and 123. Bars represent mean ± SEM; one-way ANOVA with Tukey's post hoc test. Source data are provided as a Source Data file.

harboring high heteroplasmy levels (90%) of a smaller 100-bp mtDNA deletion (called *mpt1*,[12]) within the ND1 gene (Fig. 1b) also induced miR-71 but not miR-35 (Fig. 1f, Extended Data Fig. 1d). Similarly, miR-71 but not miR-35 was induced in animals homozygous for a mutation in the proof-reading domain of the mitochondrial DNA polymerase γ (*polg-1(srh1)*) (Fig. 1g, Extended Data Fig. 1d), which causes increased levels of de novo mtDNA mutations while also reducing mtDNA copy number[52]. Finally, pharmacological inhibition of complex IV of the mitochondrial respiratory chain with sodium azide induced both miR-71 and miR-35 (Fig. 1h, Extended Data Fig. 1e). Because miR-71 was consistently induced by diverse forms of mtDNA damage, we focused on its role during mitochondrial dysfunction. Importantly, unlike mitochondrial stress, neither cytoplasmic nor endoplasmic reticulum stress induced miR-71 (Extended Data Fig. 1f-k), suggesting that it forms a small non-coding RNA response to stress originating from mitochondrial dysfunction.

## miR-71 counteracts muscle cell dysfunction caused by mtDSBs

To determine whether an increase in the abundance of miR-71 is protective against cellular dysfunction caused by mtDNA damage, we tested whether miR-71 overexpression reduced the effects of mtDSBs on muscle cells. Animals expressing *myo-3*p::mtDSBs displayed disruptions in the alignment of actin filaments that form sarcomeres (Fig. 1i, j), the basic contractile units of muscle cells. The animals also exhibited severe decreases in their rates of muscle-driven body bends (Fig. 1k) after they were forced to initiate locomotion by being placed into liquid medium, a measure of maximal muscle functional capacity. Increasing miR-71 levels six-fold via expression of the transgene *mir-71*p::miR-71 (Extended Data Fig. 1l)[46] mitigated the disruption to actin filaments (Fig. 1i, j) as well as the muscle dysfunction (Fig. 1k) caused by muscle-specific mtDSBs.

In animals carrying *mir-71*p::GFP[53], we observed fluorescence in multiple tissues including the intestine, neurons, pharynx, vulval muscle, and body wall muscles (Extended Data Fig. 1m), suggesting, as previously described[45,46], that miR-71 is expressed ubiquitously. Moreover, miR-71 is loaded into Argonaute silencing complexes in most major tissue types, including the body wall muscles, intestine, and neurons, suggesting widespread activity[54]. To determine whether miR-71 acts within muscle cells to protect against mtDSBs, we generated animals overexpressing miR-71 selectively in this tissue (*myo-3*p::miR-71) in backgrounds in which the endogenous *mir-71* gene was deleted (*mir-71(n4115)*). Interestingly, deletion of *mir-71* did not exacerbate muscle dysfunction during mtDSBs (Extended Data Fig. 1n), possibly due to redundancy in the system. However, we found that selective overexpression of miR-71 in muscle cells was sufficient to promote muscle recovery in the presence of muscle-specific mtDSBs (Fig. 1l). Selective overexpression of miR-71 in either the intestine (*ges-1*p::miR-71) or neurons (*rgef-1*p::miR-71) did not reduce the vulnerability of muscle cells towards mtDSBs (Fig. 1l). Collectively, these results suggest that upregulation of miR-71 in response to mtDNA damage is protective of tissue function in a cell-autonomous manner.

## miR-71 targets the UPR^mt regulator *dve-1*

Nearly all animal miRNAs regulate their target mRNAs by binding to short recognition sequences in their 3' untranslated regions (UTRs) through complementarity, leading to mRNA degradation or translational repression[55]. 3'UTRs of target mRNAs often contain multiple recognition sites with either perfect or imperfect miRNA complementation. To identify targets of miR-71 (seed sequence: AGAAAG), the regulation of which could reduce cellular vulnerability toward mitochondrial damage, we used TargetScan[56] to predict potential miR-71 3'UTR binding sites across the *C. elegans* transcriptome. We found 399 mRNAs with putative miR-71 recognition sites (UCUUUC) which we further sorted for mitochondrial-related functions. Interestingly, five potential miR-71 targets encoded key components of pathways involved in mitochondrial quality control and metabolic signaling, including *dct-1* and *atg-2* which mediate mitochondrial autophagy (mitophagy), *dve-1* and *hsp-6*, which are involved in the UPR^mt, and *daf-2* which regulates insulin/IGF-1 signaling. To experimentally determine whether any of these genes were regulated by miR-71, we generated transgenic reporter strains where the 3'UTR of each putative target was incorporated into a fluorescent transgene and integrated as a single-copy genome insertion to ensure consistent expression (Extended Data Fig. 2a). Each *C. elegans* reporter strain, as well as a control line harboring the *tbb-2* (tubulin) 3'UTR that does not contain a predicted miR-71 binding site, were crossed to animals overexpressing miR-71 (*mir-71*p::miR-71). Similarly to the *tbb-2* 3'UTR control reporter, the 3'UTR reporters for *dct-1*, *atg-2*, *hsp-6* and *daf-2* were not suppressed by miR-71 overexpression (Extended Data Fig. 2b). However, the reporter containing the 3'UTR of *dve-1* was significantly suppressed (Fig. 2a, b), suggesting that miR-71 can negatively regulate a key transcription factor of the UPR^mt.

To confirm this interaction, we mutated the two putative miR-71 binding sequences within the *dve-1* 3'UTR reporter (Fig. 2c) and remade the transgenic strains as above. Disrupting the putative miR-71 binding sites not only increased the overall fluorescent signal of the reporter but also abolished any suppression caused by miR-71 overexpression (Fig. 2d, e). We therefore conclude that *dve-1* transcripts are true regulatory targets of miR-71, and that the predicted miR-71 binding sites within the *dve-1* 3'UTR are required for miR-71-mediated suppression. To determine whether miR-71 regulated the abundance of endogenous *dve-1* transcripts, we quantified their mRNA levels using qPCR. In the absence of mtDNA damage, we found that *dve-1* mRNA levels were not significantly affected by miR-71 overexpression (Fig. 2f). However, upon muscle-specific mtDSBs, *dve-1* transcript levels increased six-fold. Overexpression of miR-71 in this background restored *dve-1* transcript levels to basal levels (Fig. 2f), suggesting that miR-71 can post-transcriptionally attenuate *dve-1* upregulation during severe and chronic mtDNA damage by promoting the degradation of *dve-1* mRNAs. We did not identify putative miR-71 binding sites in genes encoding other UPR^mt regulators, such as the transcription factor *atfs-1* or the ubiquitin-like protein *ubl-5*, suggesting that *dve-1* is the only major UPR^mt regulator targeted by miR-71. Supporting this notion, unlike *dve-1*, we did not detect decreases in the transcript levels of *atfs-1* or *ubl-5* when miR-71 was overexpressed, while their abundance increased due to muscle-specific mtDSBs (Extended Data Fig. 2c, d).

## miR-71 is cytoprotective against excessive UPR^mt activation

As a transcriptional regulator of the UPR^mt, DVE-1 activates a wide range of genes involved in mitochondrial repair, detoxification, and stress resistance[10,57]. It is therefore surprising that miR-71 acts to suppress *dve-1* during mtDNA damage. To resolve this apparent paradox, we sought to understand how activation of the UPR^mt affects cell function during mtDSBs. Using two fluorescent mitochondrial chaperone reporters, *hsp-6*p::GFP and *hsp-60*p::GFP[9], which are commonly used to assess UPR^mt activation, we observed increased GFP fluorescence in muscle cells in response to mtDSBs (Fig. 3a-d), suggesting that UPR^mt activation is coincident with the upregulation of miR-71 during mtDNA damage stress. We hypothesized that miR-71 orchestrates negative feedback to reduce the activities of the UPR^mt post-transcriptionally, which may be beneficial when UPR^mt hyper-activation is detrimental to cellular homeostasis. Indeed, over-expression of a constitutively active version of the key UPR^mt regulator ATFS-1 (ATFS-1^nuc) selectively in muscle cells exacerbated muscle cell dysfunction in the presence of mtDSBs, indicating that chronic UPR^mt activation is detrimental during severe mitochondrial stress (Fig. 3e). Supporting the hypothesis that miR-71 mitigates the detrimental effects of excessive UPR^mt, overexpressing miR-71 significantly improved muscle function in animals expressing ATFS-1^nuc (Fig. 3e).

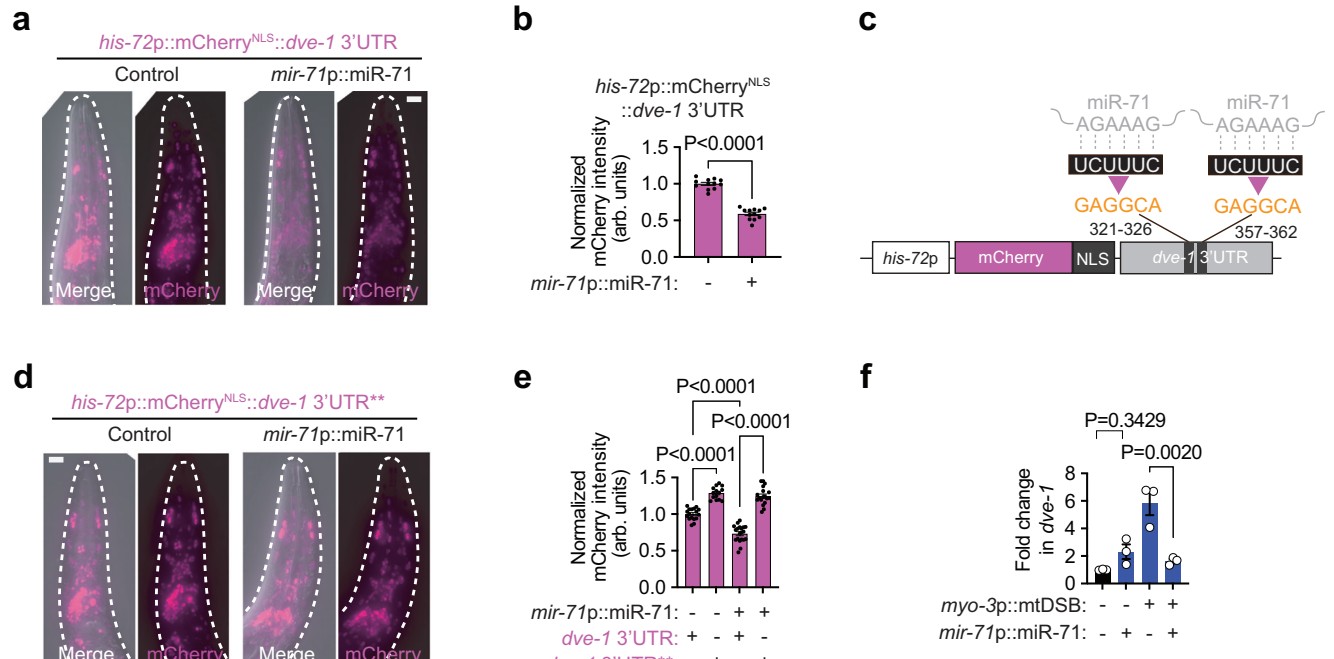

**Fig. 2 | miR-71 targets *dve-1* transcripts for degradation. a-b, (a)** representative photomicrographs of the heads of live animals (outlined by a white dashed line) and **(b)** quantification of mCherry fluorescence intensity of the *his-72*p::mCherry^NLS::*dve-1* 3'UTR reporter. Scale bar, 20 μm. Columns represent mean ± SEM; *n* = 12; two-way Student's *t* test. **c** schematic representation of the *dve-1* 3'UTR reporter annotated with the two putative sites and sequences (black boxes with white nucleotide sequences) of miR-71 seed sequence binding (gray sequences). These sites were mutated (orange sequences) to determine their relevance for miR-71-mediated regulation. **d, e, (d)** representative photomicrographs of the heads of live animals (outlined by a white dashed line) and **(e)** quantification of mCherry fluorescence intensity of the *dve-1* 3'UTR and *dve-1* 3'UTR** (seed recognition sites mutated) reporters. Scale bar, 20 μm. Columns represent mean ± SEM; *n* = 17, 20, 14 and 18; one-way ANOVA with Tukey's post hoc test. **f** qPCR analysis of *dve-1* mRNA levels. Columns represent mean ± SEM; *n* = 3 where each biological replicate is a population grown on a different plate; one-way ANOVA with Tukey's post hoc test. Source data are provided as a Source Data file.

Furthermore, several lines of evidence indicated that miR-71-mediated cytoprotection is caused by a reduction in the DVE-1-mediated UPR^mt response to mtDSBs. First, deleting *mir-71* exacerbated muscle dysfunction in animals overexpressing *dve-1* in the presence of muscle-specific mtDSBs (Fig. 3f). Second, upregulation of *dve-1* in response to mtDSBs was abolished by miR-71 overexpression (Fig. 2f). Third, *hsp-6*, a downstream effector of the UPR^mt and a DVE-1 transcriptional target was upregulated in response to mtDSBs and suppressed when miR-71 was overexpressed under its native promoter or in a muscle-specific manner (Fig. 3g). Fourth, mutating the two miR-71 binding sites within the endogenous 3'UTR of *dve-1* via genome editing significantly reduced miR-71-mediated improvements in muscle function (Fig. 3h) and muscle structure (Fig. 3i, j) in the presence of mtDSBs. And lastly, knocking down *dve-1* by RNA interference (RNAi) emulated the effects of *mir-71* overexpression, alleviating the mtDSB phenotype (Fig. 3k). Together, these results suggest that miR-71 functions to attenuate DVE-1-mediated UPR^mt hyperactivity in response to chronic mtDNA damage.

## miR-71 is regulated through the integration of ATFS-1, DAF-16 and HIF-1 activities

As our results implicated miR-71 as a negative feedback regulator of DVE-1-mediated UPR^mt, we hypothesized that *mir-71* itself must be regulated in proportion to mitochondrial stress levels in order to counteract UPR^mt hyperactivation triggered by severe or chronic damage. Indeed miR-71 abundance correlated with the extent of mtDNA defects. For instance, miR-71 levels were not significantly increased in animals harboring *uaDf5* at a heteroplasmy of 35% but increased more than nine-fold in animals with 60% *uaDf5* heteroplasmy (Fig. 1e). miR-71 abundance also

remained at normal levels in heterozygous *polg-1(srh1)* mutants but increased strongly in homozygous mutants (Fig. 1g). These results hinted at a mechanism whereby miR-71 upregulation is induced once mitochondrial stress levels surpass a certain threshold. Given that the levels of the *pre-mir-71* precursor and mature miR-71 increased by the same amount in response to mtDSBs as well as sodium azide (Fig. 4a, Extended Data Fig. 3a), we reasoned that miR-71 was regulated through transcription, rather than through miRNA biogenesis.

To understand how *mir-71* was transcriptionally upregulated in response to mitochondrial stress, we investigated the requirements of several conserved pathways which are known to act as sensors of mitochondrial and metabolic state, including ATFS-1 signaling. Our results revealed that mtDSBs promoted nuclear accumulation of ATFS-1::GFP in muscle cells (Fig. 4b, c), which alongside the induction of *hsp-6*p::GFP and *hsp-60*p::GFP (Fig. 3a–d), confirmed that mtDNA damage invokes ATFS-1-mediated UPR^mt signaling. Furthermore, ATFS-1 was required for *mir-71* upregulation in response to sodium azide, as miR-71 induction was suppressed in several *atfs-1* loss-of-function mutants, including *atfs-1(cmh15)* null animals (Fig. 4d, Extended Data Fig. 3b). However, miR-71 levels were not increased in *atfs-1(et15)* mutants, which harbor a defective mitochondrial targeting signal in ATFS-1, rendering it constitutively nuclear localized and active (Extended Data Fig. 3c). This suggested that ATFS-1 is required but not sufficient to upregulate miR-71. As such, we next investigated whether other factors contribute to *mir-71* upregulation during mitochondrial stress.

Analysis of the upstream promoter region of *mir-71* revealed putative binding sites for ATFS-1 as well as several other transcription factors that are involved in cellular metabolic and

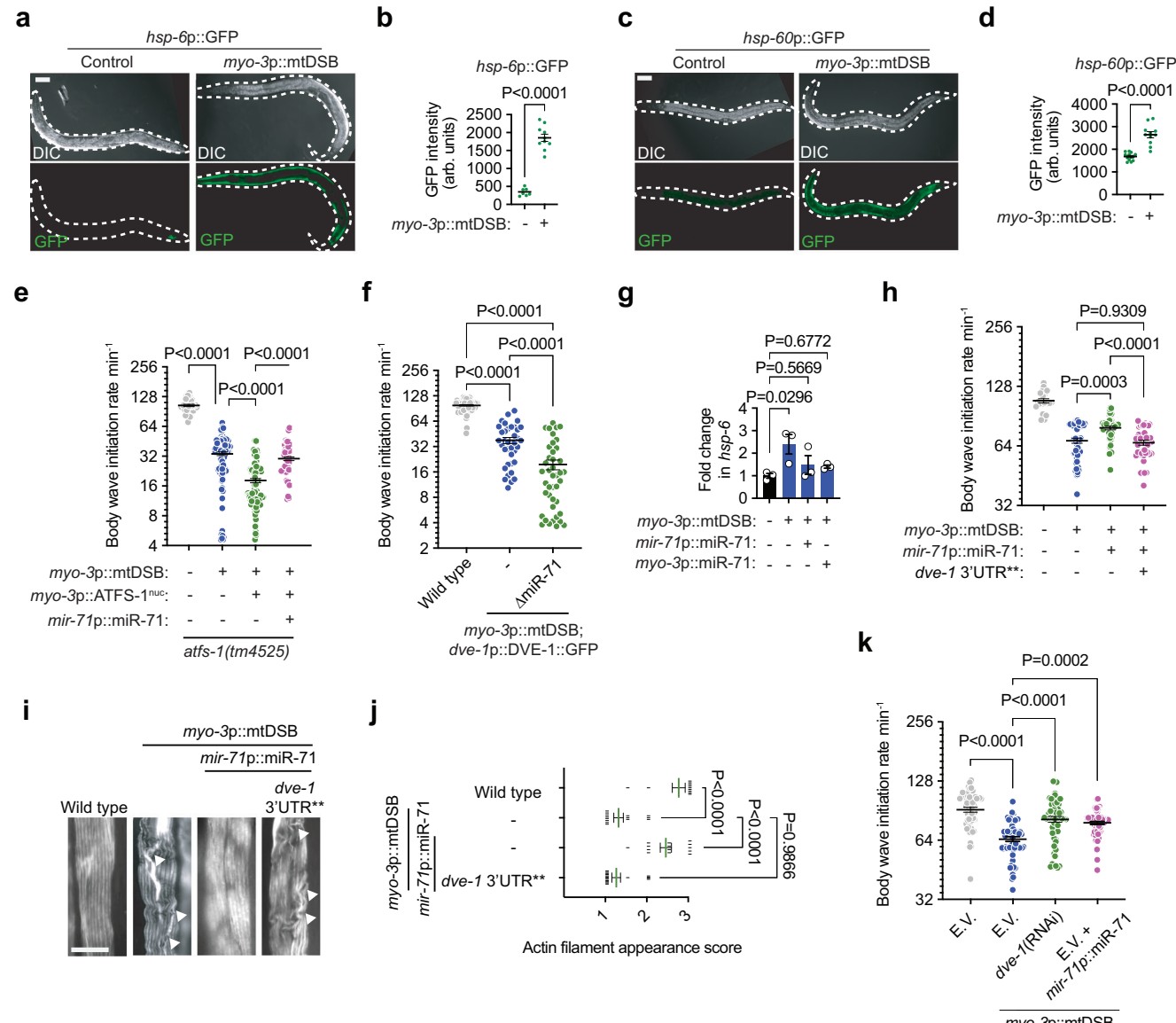

**Fig. 3 | miR-71 protects against maladaptive UPRmt hyperactivation. a-d, (a)** and **(c)** representative photomicrographs of animals (outlined by a white dashed line) and **(b)** and **(d)** quantification of GFP fluorescence intensity of the *hsp-6*p::GFP (*n* = 7 and 10) and *hsp-60*p::GFP reporters (*n* = 13 and 11). Scale bar, 50 μm. Bars represent mean ± SEM; two-way Student's *t* test. **e, f** quantification of body wave initiation rate of L4 animals placed in liquid. Bars represent mean ± SEM. For e, *n* = 36, 100, 111 and 67; for f, *n* = 63, 37 and 42; one-way ANOVA with Tukey's post hoc test. ATFS-1nuc is a constitutively nuclear-localized version of ATFS-1. **g** qPCR analysis of *hsp-6* mRNA levels. Columns represent mean ± SEM; *n* = 3 where each biological replicate is a population grown on a different plate; one-way ANOVA with Tukey's post hoc test.

**h** quantification of body wave initiation rate of L4 animals placed in liquid. Bars represent mean ± SEM; *n* = 22, 49, 49 and 48; one-way ANOVA with Tukey's post hoc test. **i** representative photomicrographs of body wall muscle tissue in L4 animals stained with phalloidin. Arrow heads indicate disruptions to actin filament structure. Scale bar, 10 μm. **j** quantification of actin filament appearance. Bars represent mean ± SEM; *n* = 11, 14, 14 and 17; one-way ANOVA with Tukey's post hoc test. **k** quantification of body wave initiation rate of L4 animals placed in liquid. Bars represent mean ± SEM; *n* = 59; one-way ANOVA with Tukey's post hoc test. For **(e–k)** WormLab automated software was used to quantify body wave initiation rates. Source data are provided as a Source Data file.

oxidative stress signaling, including the FOXO transcription factor DAF-16 and the hypoxia-inducible factor 1 (HIF-1) (Extended Data Fig. 3d). DAF-16 and HIF-1 are regulated via shuttling from the cytoplasm to the nucleus, which can occur in response to starvation and hypoxic/oxidative cellular cues, respectively, but also upon mitochondrial stress[58–60]. We found that mtDSBs invoked nuclear translocation of both DAF-16::GFP and HIF-1::GFP in muscle cells (Fig. 4b, c). Furthermore, increases in miR-71 levels caused by sodium azide exposure were abolished in *daf-16(mu86)* mutants and significantly reduced in *hif-1(ia07)* mutants (Fig. 4e, f). These results suggest that ATFS-1, DAF-16, and HIF-1 activities

are required to fully induce *mir-71* expression in response to mitochondrial stress. Consistent with this notion, *mir-71* upregulation was not further inhibited in *daf-16(mu86);atfs-1(tm4525)* or *daf-16(mu86);atfs-1(tm4525);hif-1(RNAi)* genetic backgrounds relative to individual genetic mutations in *daf-16* or *atfs-1* (Fig. 4g). Finally, constitutive DAF-16 activation caused by the *daf-2(e1370)* mutation[61] was insufficient to increase miR-71 levels and instead reduced their abundance (Extended Data Fig. 3e). In addition, transgenic expression of an undegradable and therefore constitutively active HIF-1P621G variant[62] was also unable to induce miR-71, similar to animals harboring an *atfs-1(et15)* constitutively

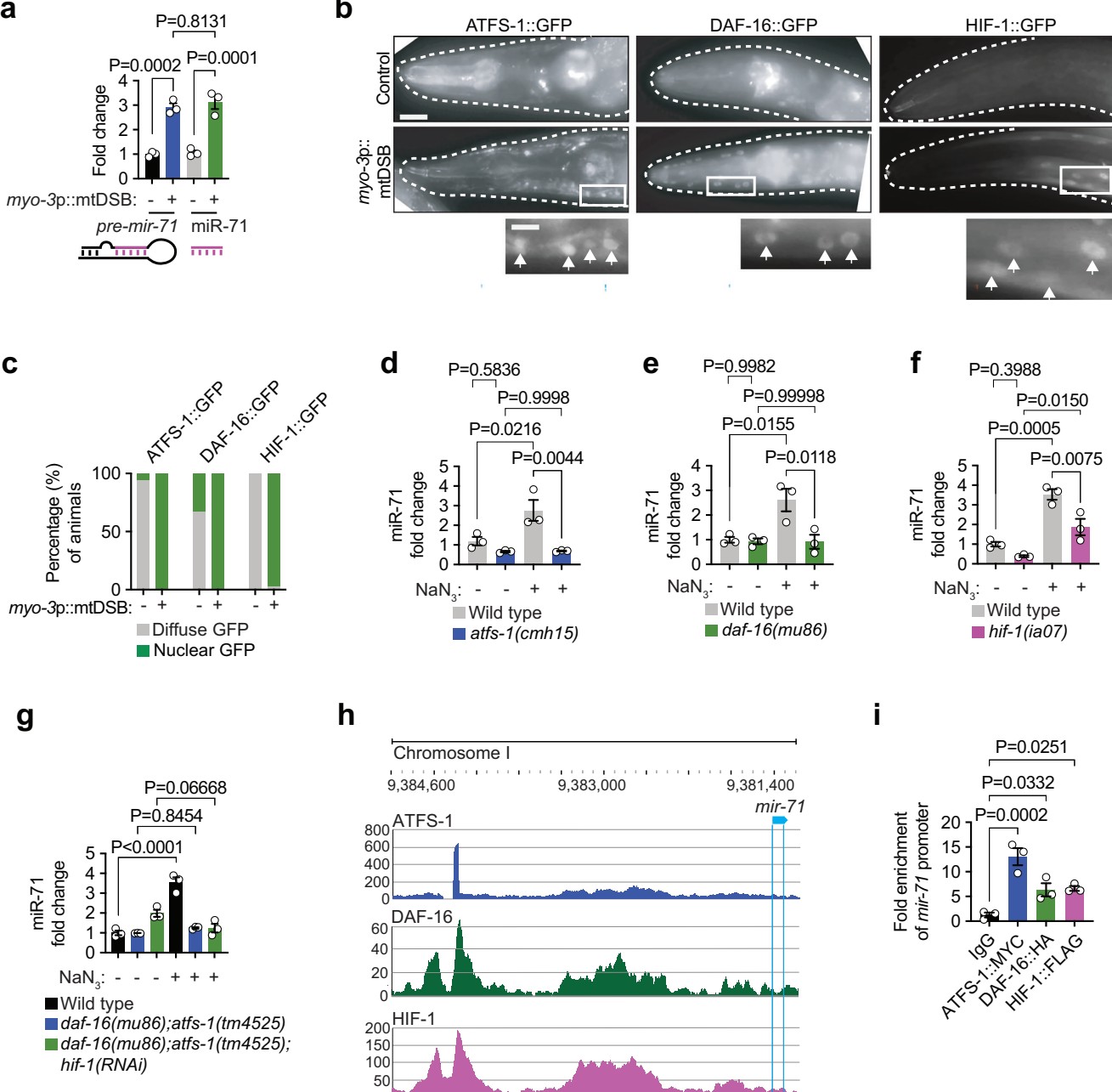

**Fig. 4 | ATFS-1, DAF-16, and HIF-1 activities are required for miR-71 upregulation during mitochondrial stress. a** qPCR analysis of *pre-mir-71* and mature miR-71 levels. Columns represent mean ± SEM; *n* = 3 where each biological replicate is a population grown on a different plate; one-way ANOVA with Tukey's post hoc test. **b** representative photomicrographs of the heads of animals (outlined by a white dashed line) expressing either ATFS-1::GFP, DAF-16::GFP or HIF-1::GFP. Scale bar, 15 µm. White box indicates enlarged area visible in the panels below, and white arrows indicate nuclei and localization of each GFP-labeled transcription factor. Scale bar, 5 µm. **c**, quantification of nuclear localization of each transcription factor.

*n* = 34, 64, 44, 56, 33, 63. **d–g** qPCR analysis of miR-71 levels. Columns represent mean ± SEM; *n* = 3; one-way ANOVA with Tukey's post hoc test. **h** chromatin immunoprecipitation (ChIP) followed by sequencing signal tracks for ATFS-1, DAF-16, and HIF-1 upstream of *mir-71*. **i** ChIP-qPCR analysis of ATFS-1::MYC, DAF-16::HA, and HIF-1::FLAG on the *mir-71* promoter sequence. Columns represent mean ± SEM; *n* = 3 where each biological replicate represent independently transfected cells; one-way ANOVA with Tukey's post hoc test. Source data are provided as a Source Data file.

active allele (Extended Data Fig. 3c and e). Moreover, in animals with double as well as triple mutant/transgene combinations that should activate two or all three pathways simultaneously, we still did not observe miR-71 upregulation (Extended Data Fig. 3e). These findings support the notion that ATFS-1, DAF-16, and HIF-1 activities are required but not sufficient to activate *mir-71* transcription, and that other pathways, yet to be identified, also contribute to miR-71 induction during mitochondrial stress.

**ATFS-1, DAF-16 and HIF-1 bind to the *mir-71* promoter sequence**

Mining of publicly available ChIP-sequencing data sets for ATFS-1 (GSE63803), DAF-16 (ENCSR946AUI), and HIF-1 (ENCSR991HIA) revealed that all three transcription factors occupied overlapping regions of the *mir-71* promoter region during stress (Fig. 4h). We confirmed this physical interaction by simultaneously expressing ATFS-1, DAF-16, and HIF-1 in HEK293T cells, and demonstrating that they each could bind to a sequence corresponding to the *mir-71*

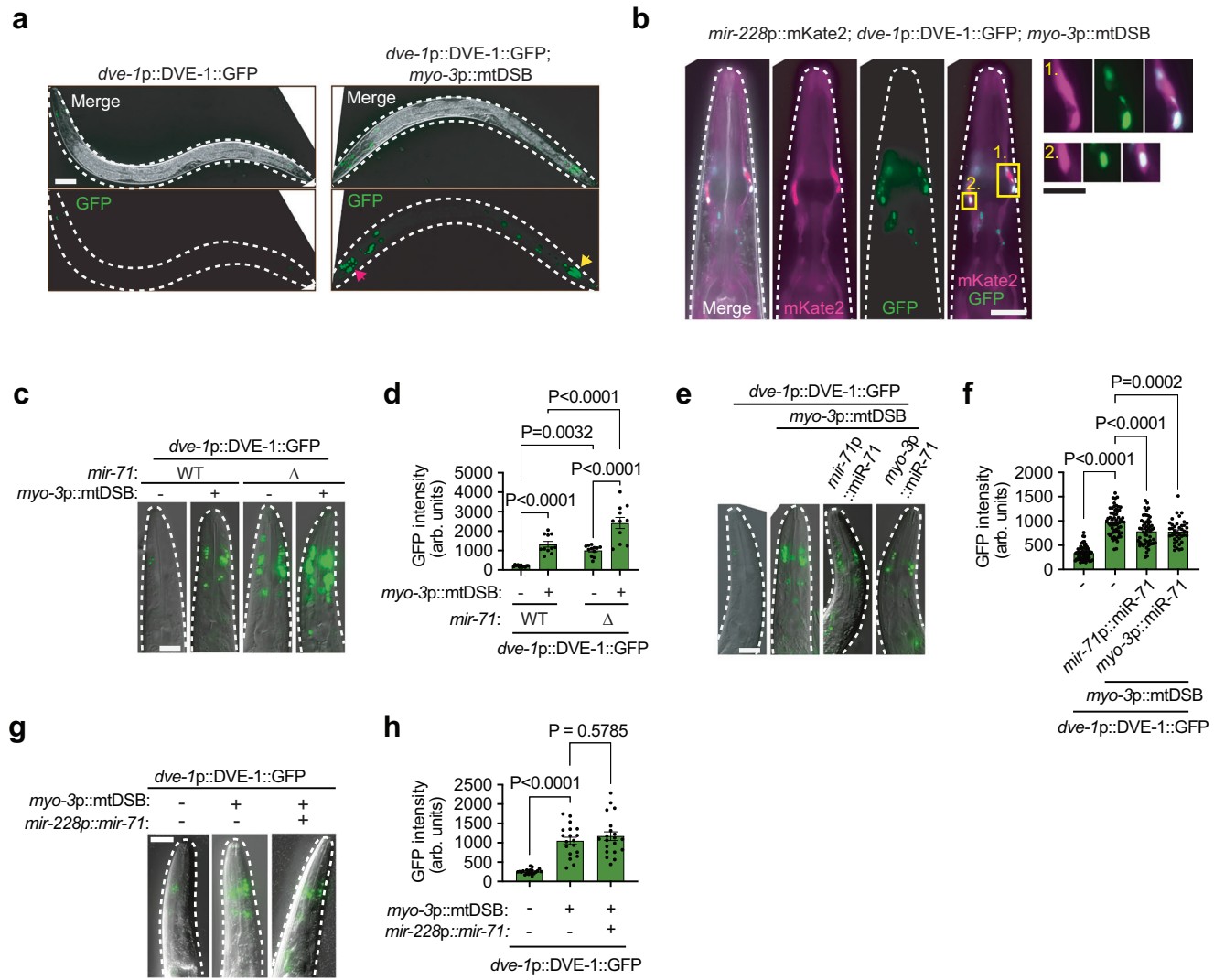

**Fig. 5 | miR-71 coordinates muscle-glia cell-non-autonomous stress signaling.**
**a** representative photomicrographs of animals (outlined by a white dashed line) expressing *dve-1*p::DVE-1::GFP. Red arrow indicates GFP signal in head cells. Yellow arrow indicates GFP signal in posterior intestinal cells. Scale bar, 50 μm.
**b** representative photomicrographs of the heads of animals (outlined by a white dashed line) expressing the glial reporter *mir-228*p::mKate2 and *dve-1*p::DVE-1::GFP. Scale bar, 20 μm. Yellow boxes correspond to enlarged images displayed on the right side that show a close-up of glial cells. Scale bar, 10 μm. **c-h**, (**c**), (**e**) and (**g**) representative photomicrographs of the heads of animals (outlined by a white dashed line) expressing *dve-1*p::DVE-1::GFP. Scale bar, 20 μm. **d-h** quantification of GFP fluorescence intensity. Columns represent mean ± SEM; for **d** n = 11, for f, n = 59, 64, 55 and 41, for h, n = 20; one-way ANOVA with Tukey's post hoc test. Source data are provided as a Source Data file.

promoter that was also co-transfected into the cells (Fig. 4i, Extended Data Fig. 3f). However, coimmunoprecipitation experiments did not reveal any physical interaction between the three transcription factors (Extended Data Fig. 3g), suggesting that they regulated *mir-71* independently. Together, these results suggest that during mtDNA damage, ATFS-1, DAF-16, and HIF-1 localize to the nucleus where they interact with the promoter of *mir-71* to coordinate its induction. This tripartite mechanism of regulation could act to restrict *mir-71* upregulation to scenarios in which mitochondrial damage is severe enough to activate all three stress signaling pathways simultaneously, such as when invoking mtDSBs.

**miR-71 suppresses cell-non-autonomous signaling of mtDNA damage from the muscle to glia cells**
We next asked whether miR-71 activity extended beyond localized suppression of *dve-1*. The coordinated regulation of mitochondrial functions across different tissues is believed to optimize the fitness of an organism. For example, the UPR^mt and changes in mitochondrial

dynamics can be non-autonomously regulated in the intestine by mitochondrial perturbation in neurons[24–28,49]. In animals expressing *dve-1*p::DVE-1::GFP, we found that muscle-specific mtDSBs induced nuclear accumulation of DVE-1::GFP in intestinal cells (Extended Data Fig. 4a) and other cells which resembled glia in the heads of animals (Fig. 5a). We therefore generated animals expressing a pan-glial fluorescent reporter (*mir-228*p::mKate2) and confirmed that glial DVE-1::GFP was activated by muscle-specific mtDSBs (Fig. 5b), suggesting that mtDNA damage within muscle cells can be communicated cell-non-autonomously to glia. Interestingly, deletion of *mir-71* increased the basal signal of nuclear DVE-1::GFP in the glia (Fig. 5c, d), suggesting that miR-71 suppressed UPR^mt activation. Furthermore, *mir-71* deletion further enhanced the glial DVE-1::GFP signal induced cell-non-autonomously by muscle-specific mtDSBs (Fig. 5c, d), suggesting that miR-71 acts to suppress signaling that induces the UPR^mt in distal cells not experiencing mitochondrial damage themselves. In support of this notion, overexpressing miR-71 (*mir-71*p::miR-71) produced the opposite effect and significantly reduced DVE-1::GFP signals in the glia

following muscle-specific mtDSBs (Fig. 5e, f). Importantly, the DVE-1::GFP transgene harbors an *unc-54* 3'UTR that does not contain predicted miR-71 binding sites, indicating that miR-71 regulates glial UPRmt activation, rather than the expression of the reporter directly.

To determine whether miR-71 functions within muscle cells to attenuate cell-non-autonomous signaling of mitochondrial damage to glial cells, we crossed the UPRmt reporter DVE-1::GFP into animals overexpressing miR-71 specifically in muscle tissue (*myo-3*p::miR-71). Remarkably, this suppressed glial UPRmt induction invoked by muscle-specific mtDSBs (Fig. 5e, f), suggesting that miR-71 in the muscle cells can dampen cell-non-autonomous UPRmt signaling. Ruling out the possibility that miR-71 itself transported from muscle-to-glia to distally silence the UPRmt, the glial signal of a miR-71 activity reporter (*his-72*p::mCherry^NLS::*dve-1* 3'UTR) was not reduced by the selective over-expression of miR-71 in muscle tissue (Extended Data Fig. 4b, c). Furthermore, we generated animals overexpressing miR-71 specifically in the glia (*mir-228*p::miR-71), which did not reduce cell-non-autonomous glial UPRmt induction (Fig. 5g, h), reinforcing the notion that miR-71 functions within muscle to suppress muscle-to-glia mitochondrial damage signaling. This activity required target recognition by miR-71 in the muscle, as overexpression of a scrambled seed-region variant (*myo-3*p::miR-71*, Extended Data Fig. 4d), which is incapable of binding target mRNAs, did not suppress glial DVE-1::GFP (Extended Data Fig. 4e, f). Interestingly, miR-71-mediated suppression of cell-non-autonomous UPRmt signaling appeared to be independent of its regulation of *dve-1*, as disruption of the miR-71 binding sites within the endogenous *dve-1* 3'UTR did not impact miR-71-mediated suppression of muscle-to-glia UPRmt activation (Extended Data Fig. 4g, h). Similarly, a *dve-1(tm4803)* mutation did not attenuate muscle-to-glia UPRmt activation (Extended Data Fig. 4i, j), further suggesting that *dve-1* and its regulation by miR-71 has no influence on muscle-to-glia UPRmt signaling. These findings collectively indicate that miR-71 acts within muscle cells to limit the propagation of a cell-non-autonomous stress signal to glia, likely by repressing one or more target transcripts other than *dve-1*.

## miR-71 suppresses peptides required for muscle-to-glia UPRmt signaling

To identify possible targets and a mechanism by which miR-71 suppressed cell-non-autonomous mitochondrial stress signaling, we performed RNA sequencing (RNA-seq) of *mir-71* deletion mutants treated with the complex IV inhibitor sodium azide to identify transcripts regulated by miR-71 during mitochondrial stress. Under non-stressed conditions, we identified 608 transcripts with increased levels in *mir-71* deletion animals (FDR < 0.05; Fig. 6a, Supplementary Data Table 2), suggesting that these genes are either directly or indirectly regulated by miR-71. In animals additionally exposed to sodium azide, we identified 2,754 transcripts elevated in *mir-71* mutants that did not overlap with basally regulated transcripts (FDR < 0.05; Fig. 6a, Supplementary Data Table 2), representing genes that are normally suppressed by miR-71 exclusively during mitochondrial stress. In *C. elegans*, non-autonomous UPRmt signaling is mediated by multiple neuro-transmitters, neuropeptides, and insulin-like peptides[24–28]. Remarkably, within the deregulated subset of transcripts, we identified a large number of mRNAs that encode neuropeptides and insulin-like peptides that are suppressed during mitochondrial stress in a miR-71-dependent manner (Fig. 6b and Extended Data Fig. 5a). This includes those previously reported to communicate mitochondrial stress systemically, including INS-27, INS-35[28], FLP-1[63] and FLP-2[27] (Fig. 6b, Extended Data Fig. 5a, Supplementary Data Table 2), suggesting that miR-71 may act to dampen cell-non-autonomous mitochondrial stress signaling via regulation of signaling peptides.

To identify a specific peptide that could mediate muscle-to-glia UPRmt activation, we performed an RNAi screen against candidates identified by RNA-seq. Of the 70 genes tested (Supplementary Data Table 3), only suppression of *nlp-52*, which encodes a neuropeptide-like protein, significantly reduced glial UPRmt activation during muscle-specific mtDSBs (Fig. 6c, d), suggesting its involvement in muscle-to-glia mitochondrial stress signaling. We confirmed this result using the *nlp-52(tm12973)* mutant, which phenocopied *nlp-52* RNAi (Fig. 6e, f). In further support, dense-core vesicle exocytosis, which is required for neuropeptide signaling and is mediated by UNC-31 (Ca^{2+}-dependent activator protein for secretion, CAPS)[25], is necessary for muscle-to-glia UPRmt signaling as RNAi of *unc-31* reduced glial DVE-1::GFP activation in animals experiencing muscle-specific mtDSBs (Extended Data Fig. 5b, c).

Next, we generated animals in which *nlp-52* was knocked-down exclusively in body wall muscle cells through the expression of complementary sense and antisense RNA sequences driven by the *myo-3* promoter that target *nlp-52* transcripts. Muscle-to-glia UPRmt signaling was almost completely abolished in these animals (Fig. 6g, h), suggesting that NLP-52 originating from the muscle is required to activate cell-non-autonomous UPRmt in the glia. Confirming this interpretation, we re-introduced a wild type copy of the *nlp-52* gene under the control of a muscle-specific promoter (*myo-3*p) into *nlp-52(tm12973)* mutants, which restored muscle-to-glia UPRmt activation (Fig. 6e, f).

## Discussion

Environmental stressors and physiological cues that affect mitochondrial function are communicated to the nucleus and distal tissues. Although these functions are important for coordinating organismal homeostasis, excessive and chronic signaling is maladaptive. Our results suggest that miR-71 is upregulated during mitochondrial stress in muscle cells, where it directly targets *dve-1* transcripts for degradation, protecting cell function against inordinate UPRmt activation. In addition, miR-71 suppresses multiple neuropeptides and insulin-like peptides, including NLP-52 which is required in muscle cells to mediate muscle-to-glia mitochondrial stress signaling. Together, our results highlight a miRNA-mediated mechanism that can simultaneously suppress both cell-autonomous and cell-non-autonomous mitochondrial stress signaling pathways during mitochondrial dysfunction (Fig. 6i).

Because miRNAs are a class of regulatory molecules that can provide immediate, reversible, and sequence-specific post-transcriptional control across multiple genetic pathways, we speculate that miR-71 enacts rapid and broad-spectrum negative feedback on mitochondrial stress signaling. In addition to *dve-1*, which we demonstrated to be a direct target of miR-71, RNA-seq revealed that *unc-31* transcripts are suppressed by miR-71 during mitochondrial stress (FDR = 0.0045; Supplementary Data Table 2). Indeed, *unc-31* has previously been shown to be a direct target of miR-71 regulation via its 3'UTR[44]. We found that other transcripts encoding neuropeptides and insulin-like peptides with demonstrated and potential roles in cell-non-autonomous UPRmt signaling are also regulated by miR-71 during mitochondrial stress. While we demonstrated that NLP-52 mediates muscle-to-glia UPRmt activation, INS-27, INS-35, and FLP-1 have previously been shown to mediate neuron-to-intestine UPRmt activation[28,63] and FLP-2 functions in a neural sub-circuit to signal cell-non-autonomous UPRmt [27]. The rich complexity of peptides regulated by miR-71 during mitochondrial stress could represent a peptide code that communicates stress between distinct cell types, enabling interpretation of the tissue incurring mitochondrial damage and specifying the cells that can receive and act on such signals.

If miR-71 mediates broad dampening of mitochondrial stress signaling, it is important to note that miR-71 is only induced during severe levels of mitochondrial stress, such as during high (60%) but not low (35%) levels of *uaDf5* heteroplasmy. This suggests that miR-71 enacts negative feedback once a certain threshold of mitochondrial dysfunction is surpassed. We found that *mir-71* is transcriptionally regulated by the funneling of at least three distinct stress pathways and

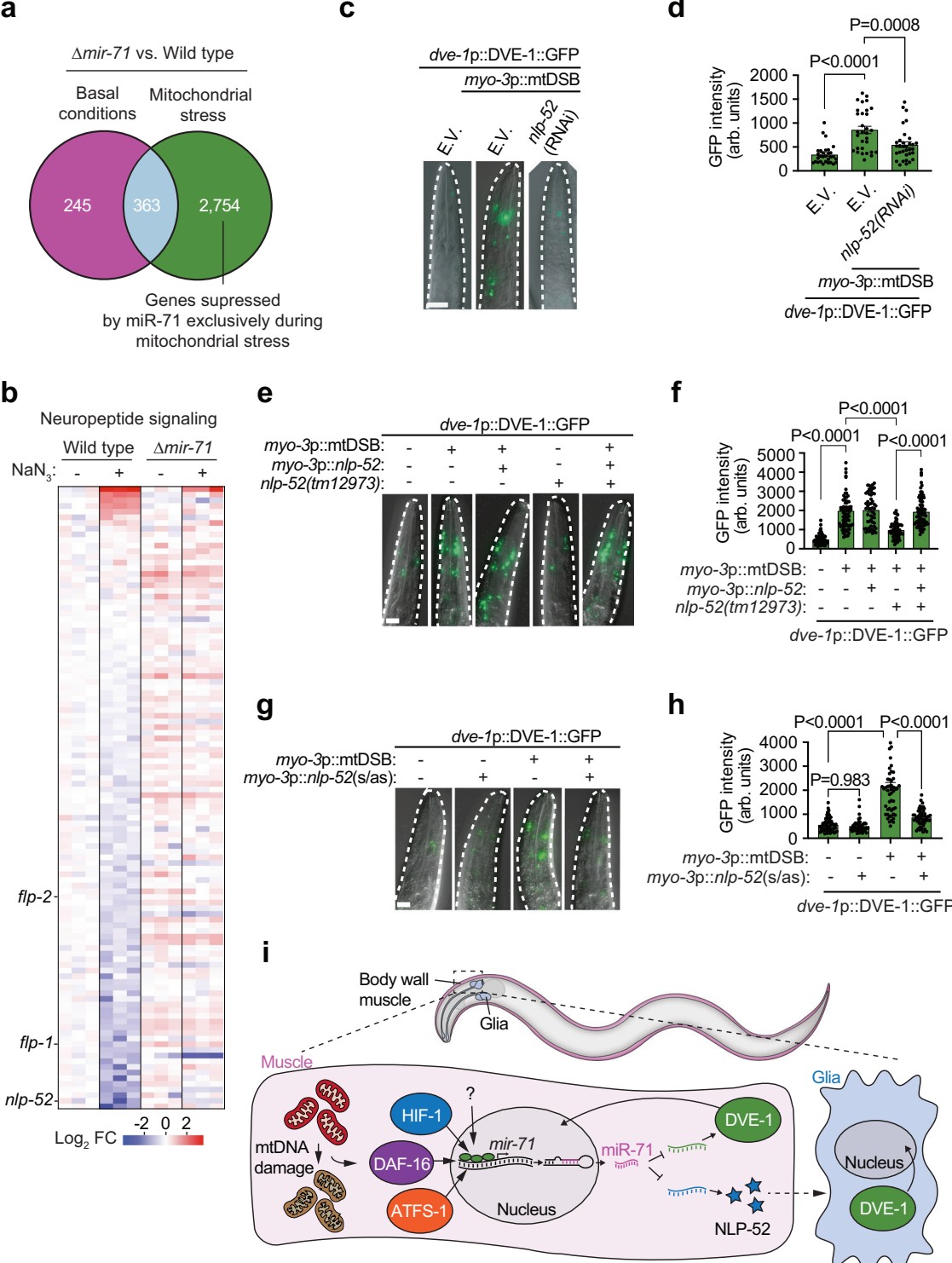

**Fig. 6 | miR-71 regulates the expression of NLP-52 which mediates cell-non-autonomous muscle-to-glia mitochondrial stress signals. a** Venn diagram of differentially expressed potential miR-71 target genes identified by RNA sequencing (RNA-seq) results of miR-71 mutant (Δmir-71) and wild-type animals. **b** heat map of RNA-seq results for transcripts involved in neuropeptide signaling. *n* = 3. **c** –**g** representative photomicrographs of the heads of animals (outlined by a white dashed line) expressing *dve-1*p::DVE-1::GFP. Scale bar, 20 μm. **d f**, and **h**, quantification of GFP fluorescence intensity. Columns represent mean ± SEM; for d, *n* = 31, 30 and 33, for f, *n* = 71, 76, 54, 63 and 76; for h, *n* = 57, 44, 54 and 50; one-way ANOVA with Tukey's post hoc test. s/as: sense/antisense. **i** proposed model of miR-71 cell-autonomous and cell-non-autonomous dampening of mitochondrial stress responses. Cell-autonomously, *mir-71* is transcriptionally upregulated during severe mitochondrial dysfunction (e.g. mtDNA damage) by the transcription factors HIF-1, DAF-16, and ATFS-1 (and possibly other pathways) where it directly targets transcripts of the UPRᵐᵗ regulator *dve-1*. miR-71 also regulates cell-non-autonomous mitochondrial stress signals by reducing the transcript levels of the neuropeptide *nlp-52*, which is required to signal muscle-to-glia mitochondrial stress. Source data are provided as a Source Data file.

their downstream transcription factors (ATFS-1, HIF-1, and DAF-16) on the promoter of *mir-71*. Because each transcription factor is required for the full miR-71 response, it is possible that this mechanism ensures that suppression of *dve-1* and cell-non-autonomous signaling is only engaged if mitochondrial stress levels are sufficient to simultaneously activate these three (and likely other) signaling pathways. Under such severe stress conditions, DVE-1-driven UPR^mt mechanisms may be deemed ineffective and instead maladaptive, particularly in the presence of high loads of mtDNA mutations and lesions that are irreparable.

In addition to levels of mitochondrial stress, the tissue-specific localization of the stress could play an important role in how signaling occurs systemically. The interpretation of mitochondrial stress emanating from muscle cells, for instance, may act as important feedback to the nervous system, which plays a critical role in the transfer of information relating to mitochondrial status across the whole animal. Glial cells were recently found to serve as upstream mediators of mitochondrial signaling across the entire organism[64]. Mitochondrial perturbations can be sensed by glia, which initiate systemic responses to promote mitochondrial stress resilience by using small clear vesicles to directly signal to neurons, which then relay the signal to the periphery via downstream neuronal mechanisms[64]. It is therefore possible that the muscle-to-glia signaling axis uncovered here could act to relay information on the status of peripheral mitochondrial health back to the nervous system, which may then communicate further instructions across the whole animal. Indeed, muscle cells may act as important sentinels of mitochondrial health due to their energetic demands that constrain metabolic flexibility. For instance, in mitochondrial myopathy caused by mtDNA defects, a mammalian integrated mitochondrial stress response (ISR^mt) proceeds in several stages that include the endocrine actions of secreted FGF21, driving signaling from muscle-to-brain and progressing the ISR^mt locally and systemically[65]. Mitochondrial myopathies have other major systemic consequences, affecting brain metabolism as well as systemic inflammation, epithelial senescence, liver steatosis, white adipose tissue abundance, and aging[65,66]. In *Drosophila*, the UPR^mt and ImpL2 (an ortholog to the human insulin-like growth factor binding protein IGFBP7) is induced by mild mitochondrial stress in muscle tissue, which upregulates mitophagy and suppresses insulin signaling throughout the whole animal, promoting longevity[67]. Therefore, mitochondrial health in muscle and the cell-non-autonomous relay of this information to other organs plays an important role in whole animal health.

Whether miR-71 acts in cells other than muscle to regulate *dve-1* and non-autonomous signaling during severe mitochondrial stress remains to be determined, although the vast number of factors it influences during systemic mitochondrial inhibition suggests that this is likely. Importantly, miR-71 has been shown to regulate multiple and diverse biological processes in a highly context-dependent manner. In the nervous system, miR-71 promotes lifespan extension by facilitating the activation of *daf-16* in the intestine upon germline removal[46], as well as through regulation of the transcription factor PHA-4/FOXA in response to dietary restriction[43]. miR-71 was also shown to mediate a direct link between food perception and the inhibition of the Toll-receptor-domain protein TIR-1 within the AWC olfactory neuron to control organismal aging and proteostasis in the intestine[47]. These neuronal functions demonstrate convergence of miR-71 activity upon the aging process via the regulation of distinct longevity pathways. Considering the close association between mitochondrial health, proteostasis, and longevity, it would be unsurprising that miR-71 dampens hyperactive mitochondrial stress responses across tissues, thereby promoting homeostasis and organismal health. More broadly, while miR-71 directly or indirectly regulates hundreds of genes under normal conditions, during mitochondrial stress the number of transcripts that are suppressed in a miR-71-dependent manner grows into the

thousands (Fig. 6a), indicating a major role in orchestrating mitochondrial homeostasis.

How inordinate mitochondrial stress responses contribute to the pathogenesis of diseases of aging, including cancers, neuroinflammation, and neurodegeneration, is not fully understood[30–32]. It was recently proposed that chronic hyperactivation of mitochondrial stress signaling drives energetically costly gains-of-function, such as mitochondrial and protein turnover and the production and secretion of cytokines and other factors, and that this "hypermetabolic state" promotes mitochondrial disease pathophysiology by diverting energy into futile repair and recovery efforts[68]. Indeed, in *C. elegans*, UPR^mt signaling launches a broad panel of energy consuming processes, such as innate immunity and pathogen clearance programs[18], which may be irrelevant to correcting the cause of stress. Interestingly, thermogenic thyroid hormones are downregulated in patients harboring the m.3243 A > G mtDNA mutation, suggesting the presence of physiological mechanisms in mammals that curb whole animal mitochondrial stress signaling[69]. Furthermore, decreasing systemic physiological responses to mitochondrial defects in mice via GDF15 or FGF21 deletion reduces hypermetabolism and systemic inflammation, normalizes body weight, and prolongs shortened lifespans[65,66,70]. Our study suggests that miRNAs play an important and coordinated role in limiting unfettered autonomous and systemic mitochondrial stress responses, and as such could be explored as a therapeutic means of reducing mitochondrial stress hyperactivation in a broad spectrum of disorders.

## Methods
All resources used in this study are listed in Supplementary Data Table 4 to 6.

### *C. elegans* strains and maintenance
A full list of strains made for this project are listed in Table S4. Animals were maintained according to established methods[71] on Nematode Growth Medium (NGM) agar plates seeded with OP50 *E. coli* and cultured at 20 °C unless otherwise indicated. Genetic crosses were performed with the aid of SoMarker[72]. Animals with Mos-mediated single-copy insertions of transgenes (MosSCI, *foxSi* alleles) were generated as described previously[73]. Briefly, gonads of 1-day-old adults of the MosSCI acceptor strains were injected with 50 ng/µl of the corresponding plasmid (PureLink™ HiPure Plasmid Miniprep Kit, Invitrogen). After injection, animals were kept at 25 °C to enable Mos-transposase-mediated insertion, which was confirmed by PCR approximately 2 weeks after injection. Animals carrying extra-chromosomal arrays were obtained by injecting 10 ng/µl of plasmid DNA (PureLink™ Quick Plasmid Miniprep Kit, Invitrogen) together with the co-injection marker *odr-1p::dsRed* or *odr-1p::GFP* (10 ng/µl) into gonads of 1-day-old adults. Subsequent transmission was checked by following the co-injection marker.

### Constructs
A full list of plasmid constructs made for this project are listed in Table S5. Most constructs were generated with overlap extension PCR (MegaWHOP), similar to previously performed[74] using Phusion™ High-Fidelity DNA Polymerase (Thermo Scientific), T4 polynucleotide kinase (NEB), T4 DNA ligase (NEB), the restriction enzyme *Dpn*I (cat. R0176, NEB), and DH5α competent bacteria. For genomic DNA (gDNA) template used for cloning, wild-type (N2) animals were lysed for 1 h at 65 °C in Mitochondrial Lysis Buffer (MLB)[75], supplemented with 0.1 mg/ml proteinase K. gDNA was subsequently extracted by phenol:chloroform following standard procedure.

The *mir-71* coding sequence was amplified from N2 gDNA using the oligos ik481 and ik474. Standard sequences of *his-72, eft-3, myo-3, ges-1* and *rgef-1* promoters, and the *tbb-2 3'UTR*, used in constructs can be found in the plasmids pSZ33, pSZ110, pSZ32, pSZ57, pSZ59, respectively[75]. To swap *tbb-2 3'UTR* in pSZ232 with alternative 3'UTR

sequences, the *dct-1 3'UTR* was amplified with the oligos ik245 and ik246, *atg-2 3'UTR* with ik247 and ik248, *hsp-6 3'UTR* with ik249 and ik250, *daf-2 3'UTR* with ik251 and ik252, and *dve-1 3'UTR* with ik255 and ik256. The two mutations of the miR-71 binding motif in the *dve-1 3'UTR* were introduced by PCR using the oligos ik282/ik283 and ik293/ik294. The promoter region of *mir-228* was amplified from N2 gDNA using the oligos ik473 and ik472. The *mir-71 locus (mir-71p::mir-71::mir-71 3'UTR)* was amplified from N2 gDNA using the oligos ik343 and ik463. All oligonucleotide sequences are listed in Table S6.

pSZ336 *(CMVp::hif-1(P621G)::3xFLAG::P2A::atfs-1::2xMYC::T2A::daf-16::HA)* was created by assembling gBlock cDNA fragments (IDT) of *atfs-1 (NM_074114)*, *daf-16 (NM_001381205)* and *hif-1(NM_075607.8)*. *C. elegans* cDNA sequences were codon-optimized for expression in mammalian cells. HIF-1 degradation was prevented by using P621G mutation[62]. cDNA fragments were assembled using Phusion DNA polymerase (NEB), Taq DNA ligase (NEB), and T5 exonuclease (NEB) following the standard Gibson assembly protocol[76,77].

## Small RNA sequencing

Synchronized worm populations were obtained by egg pulse, which was performed by isolating 100 egg-laying adults for 2 h in three biological replicates per strain, removing adults, and then allowing progeny to hatch and develop. When the progeny reached the L4 stage, larvae were collected in 1 ml of M9 buffer with 0.01% Tween20 and washed twice. 500 µl of TRIzol was added to the pelleted animals and the samples were freeze-cracked three times in liquid nitrogen to break open the cuticle. The standard TRIzol extraction protocol was followed, and the final RNA pellet was diluted in 20 µl of nuclease-free water. Small RNA sequencing libraries were prepared using the Illumina small RNA sequencing kit with 1 µg of total RNA input. Libraries were checked for size on an Agilent 2100 Bioanalyzer system, and their concentration was measured using the Invitrogen™ Qubit™ Fluorometer. Libraries were pooled and sequenced on an Illumina NextSeq sequencer. The obtained reads were mapped to the *C. elegans* genome version ce10 and annotated to miRBase release 21. Data was thresholded to adjusted P ≤ 0.05 and a fold change of 2 for downstream analysis.

## Locomotion assay

For all locomotion assays, animals were synchronized by egg pulse by allowing 30 egg-laying adults to produce eggs for 2 h before removing them from the NGM plate and raising the progeny to the L4 stage. The L4 animals were collected in 500 µl of M9 buffer with 0.01% Tween20 and transferred to 35 mm NGM plates without food using 1 ml of M9 buffer. After one minute of free swimming (adaptation time) the plates were placed under a Nikon SMZ745T stereomicroscope with a TrueChrome IIS camera (Tucsen Photonics) at 1.5x magnification and swimming behavior was recorded for 1 min. Body bends per minute were quantified using WormLab software (MBF Bioscience). 720 frames of video were analyzed by the software and at least 250 continuously trackable frames from individual animals were used for quantification.

## Microscopy and image analysis

For imaging, animals were immobilized in 10 µl of 25 mM Tetramisole (Sigma) on a 2% agarose (dissolved in $H_2O$) pad on a glass slide and covered with a coverslip. Microphotographs were taken on a Zeiss Z2 imager microscope equipped with a Zeiss Axiocam 506 mono camera, using the ZEN 2 software (Zeiss, version 2.0.0.0). Image processing and analysis were performed with FIJI software[78].

## Real-time qRT-PCR analysis of mRNAs

Total RNA was extracted using TRIZol according to the manufacturer's instructions and dissolved in 20 – 50 µl of ultrapure water (Invitrogen), depending on RNA pellet size. 1 µl was reverse-transcribed in a volume of 20 µl containing 4 µl of 5X ProtoScript II reaction buffer (NEB), 25 µM dNTPs, 50 µM oligo dT, 50 µM random hexamers, 2 µl DTT, 20 units of ProtoScript II reverse transcriptase (NEB), and 0.25 µl of murine RNase inhibitor (NEB). The reaction was incubated at 42 °C for 1 h followed by enzyme inactivation at 65 °C for 15 min. cDNA was diluted 1:20 and real-time quantitative reverse transcriptase PCR (RT qPCR) was run on a LightCycler 480 II (Roche) in a total volume of 10 µl containing 5 µl SensiFAST SYBER No-ROX (Bioline Meridian Biosystems), 1 µl of cDNA and 500 nM of each primer at the cycling conditions of 95 °C for 3 min, followed by 45 cycles of denaturation at 95 °C for 10 s, annealing at 60 °C for 10 s, and elongation at 72 °C for 20 s. Threshold cycle (Ct) values were determined by calculating the second derivative maximum of the amplification curve for three technical triplicates for each sample. Samples were normalized to actin, a gene which did not change its expression level under the experimental conditions. Fold changes were calculated based on the $2^{-\Delta\Delta Ct}$ method.

## Real-time qRT-PCR analysis of miRNAs

Total RNA was Poly(A) tailed and reverse-transcribed as described previously[79]. Briefly, a total volume of 10 µl containing 200 ng of total RNA, 2 µl 5X ProtoScript II reaction buffer (NEB), 25 µM ATP, 25 µM dNTPs, 50 µM RT primer (ik44), 1 unit of poly(A) polymerase (Invitrogen) and 20 units of ProtoScript II reverse transcriptase (NEB) were incubated at 42 °C for 1 h, followed by enzyme inactivation at 95 °C for 5 min. MiRNA-specific primers were designed as previously described[79]. cDNA was diluted 1:5 and RT qPCR was run on a LightCycler 480 II (Roche) in a total volume of 10 µl containing 5 µl SensiFAST SYBER No-ROX (Bioline Meridian Biosystems), 1 µl of cDNA and 250 nM of each primer primer (miR-35-5p was amplified with ik16/ik17, miR-71-5p with ik38/39 and miR-59-3p with ik48/49) at the cycling conditions of 95 °C for 3 min, followed by 45 cycles of denaturation at 95 °C for 10 s, annealing at 60 °C for 10 s, and elongation at 72 °C for 20 s. Threshold cycle (Ct) values were determined by calculating the second derivative maximum of the amplification curve for three technical triplicates for each sample. Samples were normalized to miR-59, a miRNA which did not change its expression level under the experimental conditions. Fold changes were calculated based on the $2^{-\Delta\Delta Ct}$ method.

## RNA sequencing

As above, synchronized worm populations were obtained by egg pulse, which was performed by isolating 100 egg-laying adults for 2 h in three biological replicates per strain, removing adults, and then allowing progeny to hatch and develop. When the progeny reached the L4 stage, larvae were collected in 1 ml of M9 buffer with 0.01% Tween20 and washed twice. 500 µl of TRIzol was added to the pelleted animals and the samples were freeze-cracked three times in liquid nitrogen. The standard TRIzol extraction protocol was followed, and the final RNA pellet was diluted in 20 µl of nuclease-free water. Sequencing library preparation, sequencing and data analysis were carried out by Azenta (https://www.azenta.com/). 1 µg total RNA was used for library preparation with poly(A) mRNA isolated using Oligo(dT) beads. mRNA fragmentation was performed using divalent cations and high temperature, and cDNA was synthesized using random sequence primers. The purified double-stranded cDNA was treated to repair both ends and add a dA-tail in one reaction, followed by a T-A ligation to add adapters to both ends. Size selection of Adapter-ligated DNA was then performed using DNA Clean Beads and each sample was amplified by PCR using P5 and P7 primers and the PCR products were validated. Libraries with different indexes were then multiplexed and sequenced using a 2 × 150 paired-end (PE) configuration according to the manufacturer's instructions. In order to remove technical sequences, including adapters, PCR primers, or fragments thereof, and quality of bases lower than 20, pass filter data in FASTQ format were processed by Cutadapt (V1.9.1, phred cutoff: 20, error rate: 0.1, adapter overlap: 1 bp, min. length: 75, proportion of N:

0.1) to filter reads. The *C. elegans* reference genome sequence and gene model annotation files were downloaded from https://www.ncbi.nlm.nih.gov/assembly/GCF_000002985.6/. Hisat2 (v2.0.1) was used to index the reference genome sequence and to align filtered reads to the reference genome. FASTA format transcripts were converted from the known gff annotation file and indexed properly. Then, with the file as a reference gene file, HTSeq (v0.6.1) estimated gene and isoform expression levels from the pair-end filtered data. Differential gene expression analysis was completed using the DESeq2 Bioconductor package, with an adjusted $P$ value cutoff of $P < 0.05$ for differentially expressed genes.

## Pharmacological treatments

Sodium azide (Sigma) was dissolved in MilliQ $H_2O$ and diluted in M9 to a concentration of either 10 mM (miR-71 induction experiments) or 20 mM (*hsp-6*p::GFP experiments) and 0.2 ml of this solution was then spread evenly over 35 mm (4 ml) NGM plates seeded with *E. coli* OP50 and left to absorb into NGM to reach final concentrations of 0.5 mM and 1 mM, respectively. Larval stage L1 individuals were transferred onto these plates and were raised to L4 and then analyzed (miR-71 induction experiments). For *hsp-6*p::GFP experiments, L4 animals were placed on the sodium azide plates for 5 h and then assessed. Tunicamycin (Sigma) was dissolved into M9 buffer to a concentration of 0.4 μg/ml and 0.2 ml of this solution spread evenly over 35 mm NGM plates seeded with *E. coli* OP50 and allowed to absorb into the NGM to reach a final concentration of 0.02 μg/ml. L4 individuals were transferred onto these plates and allowed to develop for 5 h before analysis.

## HEK293T cell culture, transfection, and co-immunoprecipitation

HEK293T cells were grown in 10 cm culture dishes in Dulbecco's Modified Eagle Medium (Gibco) supplemented with 10% fetal bovine serum at 37 °C and 5% $CO_2$. When the cells reached 60-70% confluency, 10 μg of plasmid pSZ336 (and for co-transfection 5 μg of pSZ314) were transfected with X-tremeGENE™ 9 (Roche) in Opti-MEM medium (Gibco) according to the manufacturer's instructions. When cells reached full confluency, they were washed with cold PBS and lyzed in 500 μl of lysis buffer supplemented with proteinase inhibitor (Sigma) and 1 mM phenylmethylsulfonyl fluoride (PMSF) on ice for 30 min with occasional vortexing. Debris was removed by centrifugation at 12,000x$g$ at 4 °C for 15 min. An input of 60 μl was kept and 400 μl of supernatant were diluted with additional 600 μl of supplemented lysis buffer and subjected to immunoprecipitation with either 20 μl anti MYC-beads (Pierce, 88842), 50 μl of anti-FLAG beads (Pierce, A36797), or 25 μl of anti-HA beads (Pierce, 88857) and incubated at 4 °C overnight on a rotator. The beads were collected with a magnetic rack and washed 3 times with ice-cold 1X PBS and 3 times with ice-cold RIPA buffer (25 mM Tris-HCl pH 7.4, 1% NP-40, 0.5% sodium deoxycholate, 0.1% SDS) containing 300 mM NaCl, followed by two more washes with ice-cold RIPA buffer containing 1 M NaCl. The washed beads were resuspended in 60 μl of lysis buffer and 20 μl of 4X western blot sample buffer. The samples were then denatured at 95 °C for 5 min, spun at 20,000x$g$ for 10 min at room temperature and ran on a standard 10% SDS PAGE western blot. Coimmunoprecipitation was judged by detecting the individual protein tags.

## Western blot

Samples were boiled in 1X western blot sample buffer at 95 °C for 5 min and centrifuged at 20,000x$g$ for 10 min at room temperature. The samples were then run on a 10% SDS polyacrylamide gel in running buffer (25 mM Tris, 250 mM glycine, 0.1% SDS) and blotted onto a PVDF membrane in ice-cold transfer buffer (25 mM Tris, 250 mM glycine, 20% ethanol). Membranes were blocked in TBST buffer (20 mM Tris-HCl (pH 7.5), 137 mM NaCl, 0.1% Tween20) containing 5% bovine serum albumin (BSA, Moregate Biotech) and incubated with primary antibodies in TBST with 5% BSA overnight at 4 °C on a shaker. After 3 washes with TBST, blots were incubated with the respective secondary antibody in TBST at room temperature for one hour and imaged on an Odyssey CLx near-infrared fluorescence imaging system (LI-COR). The antibodies and dilutions used in this study were: anti-HA (Cell Signaling, Cat. 3724), 1:5,000; anti-FLAG (Sigma-Aldrich, Cat. F3165), 1:5,000; anti-Myc (Invitrogen, Cat. PA1-981), 1:5,000; IRDye 680RD goat anti-mouse IgG (LI-COR, Cat. 68070), 1:20,000; and IRDye 800CW goat anti-rabbit IgG (LI-COR, Cat. 32211), 1:20,000.

## Chromatin immunoprecipitation

HEK293T cells were transfected as described above, with each cell culture dish being treated as a biological replicate and three replicates being used in the study. After 24 h, cells were washed 3 times with cold PBS and then fixed for 10 min at room temperature in 1% PFA in PBS in a total volume of 10 ml on an orbital shaker (20 rpm), after which, 1 ml of 2.5 M glycine was added to quench the fixation reaction and incubated for another 10 min. Cells were then washed 3 times with PBS and scraped off the culture dish in 1 ml of RIPA buffer (10 mM Tris-HCl pH 8.0, 0.1% SDS, 1% Triton X-100, 0.1% sodium deoxycholate, 1 mM EDTA, 0.15 M NaCl) supplemented with protease inhibitor (Sigma) and 1 mM PMSF, and incubated on ice for 15 min with occasional vortexing. Samples were then sonicated using a Covaris (M220 Focused-Ultrasonicator) for 450 s at 25 W, CBP 200, DF 20%. After centrifugation samples were spun at 10,000x$g$ for 10 min at 4 °C. Supernatant was collected and 1% input were taken. The rest of the sample was split into three tubes and antibody-conjugated beads were added: 20 μl of anti MYC-beads (Pierce, 88842), 50 μl of anti-FLAG beads (Pierce, A36797), or 25 μl of anti-HA beads (Pierce, 88857). The samples were then incubated for 2 h at 4 °C, rotating (12 rpm). Magnetic beads were collected on a magnetic stand (BioRad) and washed 1X with 0.5 ml TSE-150 (0.1% SDS, 1% Triton X-100, 2 mM EDTA, 20 mM Tris-HCl pH 8.1, 150 mM NaCl), 2X with 0.5 ml TSE-500 (TSE with 500 mM NaCl), 1X with 0.5 TSE-1M (TSE with 1 M NaCl), 1X with 0.5 ml LiCl (250 mM LiCl, 1% NP40, 1% deoxycholate, 1 mM EDTA, 10 mM Tris-HCl pH 8.1), and 2X with 0.5 ml TE (10 mM Tris pH 8.0, 1 mM EDTA). Then, 2×0.2 ml of elution buffer (1% SDS, 0.1 M NaHCO$_3$) was added to the beads with incubation for 15 min on a rotator at room temperature. After adding 20 μl of 5 M NaCl, cross-links were reversed over night at 65 °C. The next day, 10 μl of 0.5 M EDTA, 20 μl of 1 M Tris-HCL pH 6.5, and 5 μl of proteinase K (10 mg/ml) were added and incubated for another hour at 65 °C. DNA was then extracted by standard phenol-chloroform extraction and the pellet was diluted in 50 μl of MilliQ $H_2O$. The presence of amplicons of the *mir-71* promoter was tested for by qPCR using the oligos ik393 and ik394 (see Table S6) as described above. Fold enrichment was obtained by comparing the amount of target sequence measured in the IP isolate to the amount measured in the nonspecific IgG control isolate.

## RNA interference (RNAi)

RNAi was performed in a similar manner to that previously described[80]. Briefly, bacteria harboring sequenced-verified RNAi feeding vectors were obtained from the Ahringer library. IPTG plates were made by adding 238 μl of Luria broth (LB) medium, 8 μl of 1 M IPTG (Promega), and 4 μl of ampicillin (100 mg/ml) to the surface of 35 mm NGM plates and allowed to absorb into the NGM and dry. 200 μl of the RNAi bacterial culture (OD 0.4-0.7) was seeded onto these plates, which were subsequently kept in the dark at room temperature to grow a lawn for 24 h. Eggs were seeded onto the RNAi plates and animals raised to the L4 stage and analyzed. To generate strains expressing sense and antisense sequences targeting *nlp-52* in muscle cells, the *nlp-52* cDNA lacking start and stop codons was cloned into a MosSCI vector under the control of the muscle-specific *myo-3* promoter and used to generate transgenic animals. The *nlp-52* sequences were confirmed to have no off-target activity with DsCheck[81].

### RNAi by injection

RNAi by injection was performed as previously described[82]. Briefly, DNA from the *hif-1* RNAi feeding clone was extracted with the Invitrogen HighPure kit. Five individual 50 µl PCR reactions were set up with 100 ng of plasmid using the T7 primer and Phusion polymerase. All reactions were run on separate lanes on a 1% agarose gel and the bands running at 1,200 bp were cut and DNA was extracted using the Monarch gel purification kit according to the manufacturer's protocol. 4 µg of the purified (pooled) DNA was transcribed into RNA with T7 RNA polymerase and rNTPs for 5 h at 37 °C. RNA was purified by standard TRIzol extraction (500 µl per 100 µl transcription reaction), and the pellet was resuspended in 50 µl of nuclease-free water. 1-day-old adults were injected into the germline with the purified RNA and subsequent progeny were raised to L4 for analysis.

### Phalloidin staining

Phalloidin staining was performed as previously described[83]. Briefly, 50 L4 animals were collected and washed in M9 buffer and snap-frozen in liquid nitrogen. After thawing, all remaining liquid was removed with a SpeedVac vacuum concentrator. Dried animals were then fixed with ice-cold acetone and dried under a fume hood. 5 µl of Phalloidin–Atto 565 (Sigma) per tube was used for staining in a total volume 20 µl for 30 min in the dark. After 3 washes in M9 buffer with Tween, worms were mounted on agar pads and imaged as described above. The average number of lesions per cell was calculated for each animal. A health score was assigned to each individual, with a maximum score of 3 referring to all muscle cells showing perfect striation, 2.5 being similar but with small imperfections, 2 showing some lesions, and 1 referring to breaks in muscle fibers in all or most cells.

### Statistical analysis

For experiments comparing two conditions, data were analyzed using unpaired two-way Student's t-tests. For experiments with more than two conditions, data were analyzed using a one-way ANOVA with Tukey's or Šídák's multiple comparison tests. Statistical tests, $n$ values, and replicate numbers are noted in the figure legends. Most statistical analyses were performed in GraphPad Prism (version 10.2.0).

### Reporting summary

Further information on research design is available in the Nature Portfolio Reporting Summary linked to this article.

## Data availability

Original data used in each main and Supplementary Fig. has been provided in the data source file. Additional graphical and tabulated supplementary material has been provided in the supplementary documents. This paper analyzes existing, publicly available data. The accession numbers for these datasets are listed in the Materials and Methods section. RNA sequencing data generated in this study has been deposited to NCBI Gene Expression Omnibus under accession number GSE310750. Small RNA sequencing has been deposited to NCBI Sequence Read Archive under accession number PRJNA1354193. Source data are provided with this paper.

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

## Acknowledgements

We thank colleagues of the Zuryn laboratory for discussions and comments. The authors also thank Rowan Tweedale and Massimo Hilliard for comments on the manuscript. This work was supported by NHMRC grants GNT1128381, GNT1162553 and GNT2010813 (to S.Z.), ARC Discovery grant DP200101630 (to S.Z.), CJCADR Flagship grant (to S.Z.), an ARC Future Fellowship and Stafford Fox Senior Research Fellowship (to S.Z.), a DFG postdoctoral fellowship (to I.K.), University of Queensland International Scholarships (C.Y.D. and R.S.Y.L.), and Australian Government Research Training Program Scholarship (D.C. and A.P.).

## Author contributions

I.K. carried out most experiments. G.C.C.H., A.H., C.Y.D., D.C., A.A., R.S.Y.L. and A.P. contributed some experiments. I.K. and S.Z. conceived, designed, and interpreted the experiments and wrote the manuscript, with help from the other authors.

## Competing interests

The authors declare no competing interests.
