## [Peer Review file · Nature Communications]

miR-71 promotes muscle function during mitochondrial dysfunction by suppressing maladaptive UPR^{mt} signaling

Corresponding Author: Dr Steven Zuryn

Version 0:

Reviewer comments:

Reviewer #1

(Remarks to the Author)

This study reveals that miR-71 protects against mtDNA damage in *Caenorhabditis elegans* by interfering with mitochondrial stress signaling, thereby extending tissue health and function. miR-71 is induced in muscle cells by DAF-16, HIF-1, and ATFS-1. During severe mitochondrial damage, miR-71 restores myofibril structure and animal motility by directly inhibiting the overactivation of DVE-1. Additionally, miR-71 reduces the transmission of mitochondrial stress signals from muscles to glial cells.

The mitochondrial unfolded protein response (UPR^{mt}) is crucial for alleviating mitochondrial burden and promoting recovery, and the study highlights its potential adaptive failure in cases of severe or chronic damage. The research reveals a signaling axis between muscle and glial cells, emphasizing the importance of muscle cells in conveying mitochondrial health status information.

There are some questions that the authors need to address.

1, In the case of mtDSB, miR-71 promotes the recovery of muscle cell function by reducing DVE-1. Can directly knocking down DVE-1 restore muscle function? Additionally, could other UPR^{mt} inhibitors also restore muscle function?

2, The regulation of miR-71 expression by the genes ATFS-1, DAF-16, and HIF-1 cannot be understood through knockdown experiments alone; overexpression studies are also required. For example, in ATFS-1 loss-of-function *C. elegans*, does the overexpression of DAF-16 and HIF-1 affect miR-71 expression?

3, After mitochondrial stress is generated in muscle cells, it is transmitted to glial cells. Can the stress generated in glial cells regulate muscle cells in reverse and reduce muscle cell damage? In the case of mtDSB, if UPR^{mt} is reduced only in glial cells without reducing it in muscle cells, will this alleviate the functional damage of muscle cells, or will it increase the damage to muscle cells?

Minor issue:

The abbreviation mtDSB requires the full name.

Reviewer #2

(Remarks to the Author)

In this manuscript, the authors performed RNA sequencing in genetically engineered muscle-specific mtDSBs *C. elegans* backgrounds and identified 24 miRNAs that were significantly upregulated, including the miR-35 family of miRNAs and miR-71. Overexpression of miR-71 in muscular tissue is protective against cellular dysfunction caused by mtDSBs cell-autonomously. However, the deletion of miR-71 did not exacerbate muscle dysfunction during mtDSBs. The authors used TargetScan to predict the targets of miR-71 and found that *dve-1* can be targeted for degradation. Meanwhile, they found that ATFS-1, DAF-16, and HIF-1 can bind the upstream promoter region of miR-71. Beyond the cell-autonomous effect, the authors also found that miR-71 can suppress cell-non-autonomous signaling of mtDNA damage from the muscle to glia cells through the regulation of *unc-31* to suppress *ins-11* and *nlp-52*, alleviating the activation of UPR^{mt}. The authors concluded the cell-autonomous and cell-non-autonomous functions of miR-71 in regulating maladaptive UPR^{mt} signaling. While the manuscript presents interesting findings on the role of miR-71 in regulating UPR^{mt} signaling, several critical aspects

diminish its overall significance and depth of mechanistic insight:

Main points:

The study suggests that miR-71 targets DVE-1 and regulates UPRmt activation, but it lacks rigorous evidence to substantiate this claim. The authors primarily rely on miR-71 overexpression data without providing comprehensive analyses using miR-71 mutants or *dve-1* 3'UTR mutants under muscle-specific mtDSBs stress. This omission limits the depth of mechanistic understanding and raises questions about the robustness of the proposed pathway.

Additionally, the finding that mutation in *mir-71* does not exacerbate muscle dysfunction during mtDSBs is contradictory to the proposed protective role of miR-71. Further experiments to examine muscle dysfunction in *mir-71* mutants without mitochondrial stress would clarify the baseline effects of miR-71 and provide a more complete mechanistic insight.

The role of miR-71 in UPRmt regulation needs to be contextualized within the broader framework of known UPRmt pathways. The mainstream UPRmt regulation primarily involves ATFS-1, DVE-1, and UBL-5, which orchestrate a well-characterized response to mitochondrial stress. The authors should compare the impact of miR-71-mediated regulation to these established pathways, demonstrating whether miR-71 plays a redundant, synergistic, or distinct role.

The authors' finding that ATFS-1, DAF-16, and HIF-1 regulate miR-71 expression under mtDSBs stress is intriguing, but the functional relevance of this regulation remains unclear. How do these factors integrate with the canonical UPRmt pathway? Is miR-71 expression a downstream consequence or a parallel regulatory mechanism? Addressing these questions is crucial to establish the significance of miR-71 in the broader context of UPRmt regulation.

The manuscript proposes that miR-71 exerts protective effects without binding to its endogenous 3'UTR of *dve-1p::DVE-1::GFP* (The reporter was constructed in a vector containing *unc-54* 3'UTR). This paradoxical observation needs to be resolved through additional experiments. The authors should explore alternative mechanisms by which miR-71 might influence DVE-1 or other components of the UPRmt pathway.

The proposed cell-non-autonomous effects of miR-71 in suppressing signaling from muscle to glial cells are intriguing but underexplored. The roles of *ins-11* and *nlp-52* in this process should be validated through RNAi, tissue-specific knockout or knockdown, and tissue-specific rescue assays to understand their function in muscle and UPRmt signaling comprehensively.

In summary, the study in its current form lacks the significance and depth required to advance our understanding of UPRmt regulation. Addressing these critical points would enhance the mechanistic insight and position of miR-71 within the established landscape of UPRmt regulatory pathways.

Reviewer #3

(Remarks to the Author)

The study by Kirmes et al. describes the identification of miR-71 in muscles in response to severe mitochondrial stress and activation of DAF-16, HIF-1 and ATFS-1 and leads to degradation of DVE-1 to alleviate overactivation of the UPRmt. Additionally, to the impact of miR-71 in the muscle, they propose that miR-71 leads to reduced secretion of neuro and insulin-like peptides which act to suppress muscle to glia axis of mitochondrial stress. Therefore, they suggest that miR-71 is a novel target of the UPRmt, that act to prevent its overactivation and leads to local and systemic beneficial effects.

Overall, the introduction is well written and provide appropriate background and relevant literature. The inclusion of a nuclease dead control in the original screen is a strong control, data supporting the impact of miR71 following mitochondrial damage in muscle is strong, the results linking miR71 to the regulation of DVE1 in figure 2 is convincing and involves proper controls, same applies to all figures notably the control in figure 5 for regulation of UPRmt rather than reporter directly, that miR71 does not spread itself across tissues. Therefore, the study present compelling results and is very well designed.

While the non cell-autonomous effect of mitochondrial stress in one tissue affecting another is already known, the novelty here is to add miR71 and the fact that the strength of signal is important.

Critiques:

1- miR-35 is activated significantly more than miR-71, miR246 seems activated at similar level to miR71, several stressors also activates mir35 but *polG* represses it, which is interested. Is there a specificity within the miR35 family? the focus on miR-71 or lack of analysis of miR-35 and miR246 needs to be justified in a stronger manner, especially that the group have published in miR71 previously and therefore the choice of focusing on this particular miR come across as purely biased. The same applies to the focus on the peptides already known to regulate non-cell autonomous communication between tissues following mitochondrial stress in the screen in figure 6.

2- Overexpression of miR71 increases *daf2* significantly the receptor of insulin in muscle but this is not discussed, how it relates to the finding of insulin-like peptide in figure 6 (Extended data Fig.2).

3- Choice of sodium azide for RNAseq is unfortunate and detract from main finding using a mtDNA specific DNA damage. While some validation is provided in fig. 6d and e, a more comparison of the gene expression in panel b and c following muscle and mitochondrial specific DNA damage should be provided.

Version 1:

Reviewer comments:

Reviewer #1

(Remarks to the Author)

The authors answered my questions.

Reviewer #2

(Remarks to the Author)

The revised manuscript appropriately responded to the reviewer's comments. The additional results strengthened the author's conclusion.

Reviewer #3

(Remarks to the Author)

The authors have addressed previous concerns and provided appropriate clarifications.

miR-71 promotes muscle function during mitochondrial dysfunction by suppressing maladaptive UPR^{mt} signaling.

New figures are referred to in the text below and textual changes to the manuscript are coloured in red in the revised manuscript file.

Reviewer #1 (Remarks to the Author):

This study reveals that miR-71 protects against mtDNA damage in *Caenorhabditis elegans* by interfering with mitochondrial stress signaling, thereby extending tissue health and function. miR-71 is induced in muscle cells by DAF-16, HIF-1, and ATFS-1. During severe mitochondrial damage, miR-71 restores myofibril structure and animal motility by directly inhibiting the overactivation of DVE-1. Additionally, miR-71 reduces the transmission of mitochondrial stress signals from muscles to glial cells.

The mitochondrial unfolded protein response (UPR^{mt}) is crucial for alleviating mitochondrial burden and promoting recovery, and the study highlights its potential adaptive failure in cases of severe or chronic damage. The research reveals a signaling axis between muscle and glial cells, emphasizing the importance of muscle cells in conveying mitochondrial health status information.

We thank Reviewer #1 for their positive comments and insightful suggestions, which we have addressed below and in the revised manuscript.

There are some questions that the authors need to address:

1, In the case of mtDSB, miR-71 promotes the recovery of muscle cell function by reducing DVE-1. Can directly knocking down DVE-1 restore muscle function? Additionally, could other UPR^{mt} inhibitors also restore muscle function?

We have now performed experiments in which we knocked down *dve-1* by targeting it with RNA interference (RNAi) and show that this helps to restore muscle function in animals with muscle-specific mtDSBs (see Fig. 3k and additions to the text in the revised manuscript). The level of muscle function improvement is similar to that obtained by miR-71 overexpression, supporting the conclusion that repression of *dve-1* improves muscle function in the context of mtDSBs. We thank the reviewer for their suggestion, which we believe further strengthens the manuscript.

Although we are not aware of any pharmacological inhibitors of the UPR^{mt}, we have previously demonstrated that loss-of-function mutations in *atfs-1* (alleles *cmh15* and *tm4919*), which also regulates the UPR^{mt}, have no beneficial effect on muscle function in the context of mtDSBs¹. Importantly, restricting ATFS-1 to the mitochondria through mutation of its nuclear localisation sequence does have a beneficial effect, which appears to be independent of its UPR^{mt} role¹.

2, The regulation of miR-71 expression by the genes ATFS-1, DAF-16, and HIF-1 cannot be understood through knockdown experiments alone; overexpression studies are also required. For example, in ATFS-1 loss-of-function C. elegans, does the overexpression of DAF-16 and HIF-1 affect miR-71 expression?

We thank the reviewer for their suggestion. Importantly, the activities of each of these transcription factors is not regulated by their levels of expression, but rather through regulation of their translocation into the nucleus via changes in either mitochondrial import efficiency (ATFS-1²), post-translational modification (DAF-16³), or active degradation (HIF-1⁴). Therefore, instead of overexpressing each gene, we used a constitutively active mutant for *atfs-1* (*et15* allele), which results in exclusive nuclear localisation of ATFS-1⁵ and a loss-of-function mutant of *daf-2* (*e1370*), which results in constitutive activation and nuclear localisation of DAF-16⁶. In addition, we used an overexpression transgene of *hif-1p::hif-1*^{P621G} that carries a mutation which renders HIF-1 undegradable by VHL proteins and is therefore active and localises to the nucleus, even in the presence of oxygen⁴. We found that enhancing the activity or expression of each factor individually (even in wild type backgrounds where each of the other two factors remain present) was insufficient to increase *mir-71* expression levels and instead reduced miR-71 transcript levels in the case of *atfs-1(et15)* and *daf-2(e1370)* (Extended Data Fig. 3e).

We next crossed these mutants/transgenic strains together to create double and triple mutant strains and measured *mir-71* expression levels. Again, we did not observe increases in *mir-71* expression in any mutant combination and instead observed a reduction in miR-71 transcripts in *daf-2(e1370);hif-1p::hif-1*^{P621G}, *atfs-1(et15);hif-1p::hif-1*^{P621G}, *atfs-1(et15);daf-2(e1370)*, and triple mutant/transgenic animals (Extended Data Fig. 3e).

Together, this new data supports the proposed mechanism that activation of ATFS-1, DAF-16, and HIF-1 individually is insufficient to induce *mir-71* expression. It further suggests that although each factor is required (Fig. 4d-f), activation of all three factors together is also insufficient and point to additional requirements for *mir-71* transcriptional regulation. It also remains possible that *daf-2(e1370)* mutants activate parallel pathways that could suppress *mir-71*. Therefore, these new results support the overall conclusions that *mir-71* transcription is regulated by the combined activities of ATFS-1, DAF-16, and HIF-1 as well as possibly by other, yet to be identified factors, during mitochondrial stress. The new results are included in the revised manuscript (Extended Data Fig. 3e) and reflected in the revised model (Fig 6i).

3, After mitochondrial stress is generated in muscle cells, it is transmitted to glial cells. Can the stress generated in glial cells regulate muscle cells in reverse and reduce muscle cell damage? In the case of mtDSB, if UPR^{mt} is reduced only in glial cells without reducing it in muscle cells, will this alleviate the functional damage of muscle cells, or will it increase the damage to muscle cells?

We understand that Reviewer #1 wishes to know the impact of the cell-non-autonomous UPR^{mt} response in the glial cells on muscle function in the context of mtDSBs. This is an interesting question and to address it, we generated new strains in which we overexpressed miR-71 exclusively in the glia (via the glial-specific *mir-228* promoter) to suppress *dve-1* activity in only these cells. We crossed these transgenic animals with strains incurring muscle-specific

mtDSBs and found no change in muscle function (see Fig. 1 below). This result suggests that glial UPR^{mt} activation does not influence muscle function affected by mtDSBs. In other words, glial stress responses do not appear to affect muscle function during stress.

Figure 1. Quantification of body wave initiation rate (using WormLab automated software analysis) of L4 animals placed in liquid. Bars represent mean \pm SEM; $n > 3$; one-way ANOVA with Tukey's post hoc test.

4, The abbreviation mtDSB requires the full name.

We have now corrected this oversight in the revised manuscript.

Reviewer #2 (Remarks to the Author):

In this manuscript, the authors performed RNA sequencing in genetically engineered muscle-specific mtDSBs *C. elegans* backgrounds and identified 24 miRNAs that were significantly upregulated, including the miR-35 family of miRNAs and miR-71. Overexpression of miR-71 in muscular tissue is protective against cellular dysfunction caused by mtDSBs cell-autonomously. However, the deletion of mir-71 did not exacerbate muscle dysfunction during mtDSBs. The authors used TargetScan to predict the targets of miR-71 and found that *dve-1* can be targeted for degradation. Meanwhile, they found that ATFS-1, DAF-16, and HIF-1 can bind the upstream promoter region of mir-71. Beyond the cell-autonomous effect, the authors also found that miR-71 can suppress cell-non-autonomous signaling of mtDNA damage from the muscle to glia cells through the regulation of *unc-31* to suppress *ins-11* and *nlp-52*, alleviating the activation of UPR^{mt}. The authors concluded the cell-autonomous and cell-non-autonomous functions of miR-71 in regulating maladaptive UPR^{mt} signaling. While the manuscript presents interesting findings on the role of miR-71 in regulating UPR^{mt} signaling, several critical aspects diminish its overall significance and depth of mechanistic insight:

We thank Reviewer #2 for their insights and have addressed their concerns below.

1, The study suggests that miR-71 targets DVE-1 and regulates UPR^{mt} activation, but it lacks rigorous evidence to substantiate this claim. The authors primarily rely on miR-71 overexpression data without providing comprehensive analyses using miR-71 mutants or *dve-1* 3'UTR mutants under muscle-specific mtDSBs stress. This omission limits the depth

of mechanistic understanding and raises questions about the robustness of the proposed pathway.

We thank the reviewer for their comment and would like to emphasise that multiple orthogonal lines of evidence provide evidence that miR-71 targets the 3'UTR of *dve-1* during mitochondrial stress, which mitigates the maladaptive UPR^{mt}. As outlined below, we have analysed miR-71 deletion mutants as well as *dve-1* 3'UTR mutants (in both transgene reporters and through CRISPR-Cas9 genome editing of the endogenous gene).

1) We generated transgenic reporter strains wherein the 3'UTR sequence of *dve-1* was incorporated into a fluorescent transgene and integrated as a single-copy genome insertion. We also generated an identical reporter strain (identical genome insertion site) with a control *tbb-2* 3'UTR which lacks any putative miR-71 binding sites. Increasing the abundance of miR-71 by six-fold (Extended Data Fig.1l), which is similar to induction levels observed during mitochondrial stress (e.g. Fig. 1d, e, h), strongly suppressed the fluorescence signal of the *dve-1* 3'UTR reporter by half ($P < 0.0001$; Fig. 2a and b) but had no effect on the signal of the *tbb-2* 3'UTR reporter ($P = 0.2847$; Extended Data Fig. 2b).

2) To test whether the putative miR-71 binding sites within the *dve-1* 3'UTR were required for miR-71-mediated silencing of the *dve-1* 3'UTR reporter used above, we mutated these sequences (Fig. 2c) and reintroduced the transgene into the same genome insertion site as the original reporter. Scrambling the two putative miR-71 binding sites completely abolished any suppression of the reporter by increased levels miR-71 (Fig. 2d and e). This result confirms that miR-71 specifically targets and suppresses *dve-1* via recognition of two binding sites in the *dve-1* 3'UTR, validating the precise prediction of TargetScan and further strengthening the conclusion that *dve-1* is a bona fide target of miR-71.

3) Demonstrating the suppressive effect of miR-71 on endogenous *dve-1* transcripts, we used qPCR to show that enhancing miR-71 levels prevented increases in *dve-1* mRNA levels caused by mtDSB stress (Fig. 2f). Not only does this further support the action of miR-71 on *dve-1* but it also demonstrates that miR-71 attenuates increases in *dve-1* transcript abundance in the context of mtDSBs.

4) We performed CRISPR-Cas9 genome editing of the endogenous *dve-1* 3'UTR to mutate the experimentally validated miR-71 binding sites from point 2 above (also see Fig. 2c-e). This abolished the ability of miR-71 to suppress muscle dysfunction (Fig. 3h) and actin filament disorganisation (Fig. 3i and j) caused by mtDSBs. This experiment is what the reviewer has recommended, and not only does it further confirm that miR-71 targets *dve-1* via recognition of its 3'UTR binding sites, but also demonstrates that miR-71-mediated targeting of *dve-1* is beneficial during muscle-specific mtDSBs.

5) Downstream targets of DVE-1, such as the UPR^{mt} effector and mitochondrial chaperone *hsp-6*, are up-regulated in response to mtDSBs, but suppressed when miR-71 levels are increased (Fig. 3g). This supports a model of miR-71-mediated suppression of DVE-1, which ameliorates the up-regulation of its direct transcriptional targets during mitochondrial stress. Importantly, increasing miR-71 levels selectively in muscle cells

recapitulates this result (Fig. 3g), indicating that miR-71 acts cell-autonomously to attenuate UPR^{mt} activation during mtDSB stress.

6) In animals with enhanced *dve-1* levels, muscle dysfunction caused by mtDSBs is further exacerbated by the deletion of *mir-71* (Fig. 3f). This indicates that animals lacking miR-71 are sensitive to mtDSB, when the UPR^{mt} is overactivated through *dve-1* overexpression.

7) Finally, deletion of *mir-71* increased the fluorescence signal of the UPR^{mt} reporter (*dve-1p::DVE-1::GFP*), which was further increased in backgrounds with muscle-specific mtDSBs (Fig. 5c and d). This suggests that miR-71 suppresses UPR^{mt} signalling.

We believe that the experimental evidence outlined above is compelling and strongly supports the model that miR-71 targets *dve-1*, thereby regulating UPR^{mt} activation. We further experimentally validate the specific sequences in the *dve-1* 3'UTR that are required for this regulation and show the phenotypic effects of this under the context of mtDSBs. This is supported by Reviewer #3's comments to the authors that **"the results linking miR71 to the regulation of DVE1 in figure 2 is convincing and involves proper controls, same applies to all figures notably the control in figure 5 for regulation of UPRmt rather than reporter directly, that miR71 does not spread itself across tissues. Therefore, the study present compelling results and is very well designed."**

While miR-71 is overexpressed in many of these experiments, so too is the fluorescent reporter used to assess miRNA suppression, which is the standard approach for defining bona fide miRNA targets *in vivo*. Importantly, the six-fold increase in abundance of miR-71 in the overexpression strain (Extended data Fig.1l) is within the range of the increase observed for the endogenous miR-71 upon exposure to various mitochondrial stressors and is therefore a physiologically relevant level of expression that matches stress-induced scenarios. Specifically, muscle-specific mtDSBs induce miR-71 by ~4-fold (Fig. 1d), and animals harbouring the *uaDf5/+* mtDNA deletion (60% heteroplasmy) or are treated with NaN₃ increase miR-71 levels by ~9-fold (Fig. 1e) and ~6-fold (Fig. 1h), respectively.

As pointed out in comment 2 by the reviewer (below), loss of *mir-71* does not exacerbate muscle dysfunction caused by mtDSB. In the manuscript, we hypothesised that this may be due to miR-71 functional redundancy, which is revealed if we enhance *dve-1* expression levels (point 6 above). As such, we capitalised on the effects of increases in miR-71 levels rather than *mir-71* deletion, as overexpression recapitulates the induction of miR-71 during mitochondrial stress.

2, Additionally, the finding that mutation in mir-71 does not exacerbate muscle dysfunction during mtDSBs is contradictory to the proposed protective role of miR-71. Further experiments to examine muscle dysfunction in mir-71 mutants without mitochondrial stress would clarify the baseline effects of miR-71 and provide a more complete mechanistic insight.

We thank the reviewer for their comment, which we have addressed above. In addition, miRNA redundancy is a significant factor in genetic studies, as the absence of a single miRNA may not lead to an observable phenotype due to compensatory mechanisms by

other miRNAs. Therefore, the implication that miR-71 is functionally redundant, in principle, does not contradict our model that miR-71 is protective against the effect of mtDSBs. It does however, support the widely reported functional redundancy of miRNAs, whereby multiple miRNAs can target individual transcripts and individual miRNAs can target multiple, distinct transcripts.

3, The role of miR-71 in UPR^{mt} regulation needs to be contextualized within the broader framework of known UPR^{mt} pathways. The mainstream UPR^{mt} regulation primarily involves ATFS-1, DVE-1, and UBL-5, which orchestrate a well-characterized response to mitochondrial stress. The authors should compare the impact of miR-71-mediated regulation to these established pathways, demonstrating whether miR-71 plays a redundant, synergistic, or distinct role.

We thank the reviewer for their suggestion. Considering that the manuscript establishes that miR-71 directly targets *dve-1*, we have focussed our response on possible interactions between miR-71 and the other two components of the UPR^{mt}, *atfs-1*, and *ubl-5*.

Unlike *dve-1*, both *atfs-1* and *ubl-5* do not have putative binding sequences for miR-71 and are therefore unlikely to be direct targets of miR-71. To confirm that this is true, we performed qPCR to determine whether the mRNA levels of each gene are reduced in animals with increased miR-71 levels. Unlike *dve-1* transcripts (Fig. 2f), we found increasing miR-71 abundance did not decrease the levels of *atfs-1* or *ubl-5* transcripts (Extended Data Fig. 2c and d), suggesting that miR-71 does not target these genes. Therefore, we conclude that miR-71 plays a distinct role by targeting only the DVE-1 branch of the UPR^{mt}. These new results are included in the manuscript.

4, The authors' finding that ATFS-1, DAF-16, and HIF-1 regulate miR-71 expression under mtDSBs stress is intriguing, but the functional relevance of this regulation remains unclear. How do these factors integrate with the canonical UPR^{mt} pathway? Is miR-71 expression a downstream consequence or a parallel regulatory mechanism? Addressing these questions is crucial to establish the significance of miR-71 in the broader context of UPR^{mt} regulation.

We thank the reviewer for their comment and important questions.

ATFS-1 is a key UPR^{mt} regulator that can translocate to the nucleus and induce the transcription of ~300-400 genes that contribute to mitochondrial homeostasis and immunity⁷. Here, we find that it is also required for *mir-71* induction during severe mitochondrial stress (Fig. 4d and Extended Data Fig. 3b). Although DAF-16 and HIF-1 are not integrated within the canonical UPR^{mt} pathway, we find that they are also required for the full transcriptional up-regulation of *mir-71* during mitochondrial stress (Fig. 4e-g). Because we have demonstrated that miR-71 targets *dve-1* (Fig. 2, 3h-j), both DAF-16 and HIF-1 appear to play a negative feedback role over the DVE-1 branch of the UPR^{mt}. We propose that under severe and chronic mitochondrial stress conditions, all three transcription factors are simultaneously activated and localise to the nucleus (Fig. 4b and c), bind to the promoter of *mir-71* (Fig. 4h and i), and up-regulate *mir-71* expression, which subsequently exerts negative feedback on DVE-1, thus limiting maladaptive UPR^{mt} activation. In other words, our model suggests that miR-71 forms a parallel regulatory mechanism that counteracts the DVE-1-mediated UPR^{mt} pathway.

Similarly to what we demonstrated, it has previously been shown that DAF-16 and HIF-1 can localise to nuclei in response to mitochondrial stress⁸⁻¹⁰. We did not observe any genetic interaction (Fig. 4d-g) or detect any physical association between ATFS-1, DAF-16, and HIF-1 (Extended Data Fig. 3f and g) that would suggest that they integrate on the same pathway and instead have proposed a mechanism in which each factor is required to regulate *mir-71* expression during mitochondrial stress in an independent manner.

Furthermore, a constitutively active *atfs-1(et15)* mutant that localises to the nucleus and activates the UPR^{mt} and various UPR^{mt} reporters (e.g. *hsp-6p::GFP*) under normal conditions⁵ is not sufficient to drive *mir-71* upregulation (Extended Data Fig. 3c). This suggests that *mir-71* itself does not fall under the canonical UPR^{mt} pathway mediated by ATFS-1. It also suggests that ATFS-1 is not downstream of DAF-16 or HIF-1 signalling in this context. Instead, our results suggest that the integration of several stress pathways activated during mitochondrial stress is required for *mir-71* regulation. Moreover, these results suggest that *mir-71* induction is not simply a downstream consequence of UPR^{mt} activation, but a negative feedback mechanism that is activated in parallel during mitochondrial stress.

5, The manuscript proposes that miR-71 exerts protective effects without binding to its endogenous 3'UTR of *dve-1p::DVE-1::GFP* (The reporter was constructed in a vector containing *unc-54* 3'UTR). This paradoxical observation needs to be resolved through additional experiments. The authors should explore alternative mechanisms by which miR-71 might influence DVE-1 or other components of the UPR^{mt} pathway.

We thank the reviewer for their comment and apologise for any confusion regarding the miR-71 mechanism.

We have proposed two distinct mechanisms (illustrated in Fig. 6i) by which miR-71 acts to dampen the UPR^{mt} response during mitochondrial stress; 1) a cell-autonomous mechanism whereby miR-71 directly binds to the 3'UTR of *dve-1* transcripts, reducing its abundance, and 2) a cell-non-autonomous mechanism in which miR-71 negatively regulates signalling peptides that are required to activate the UPR^{mt} in distal glia cells.

The *dve-1p::DVE-1::GFP::unc-54* 3'UTR transgene is strictly a reporter of UPR^{mt} activation and *not* a reporter of miR-71 activity. We used this reporter to demonstrate that miR-71 reduces the activation of the cell-non-autonomous UPR^{mt} in distal glia cells (e.g. Fig. 5). In other words, miR-71 does *not* directly regulate the *dve-1p::DVE-1::GFP::unc-54* 3'UTR transgene, but instead regulates factors required for UPR^{mt} activation in glial cells. The fact that the reporter transgene does not have a *dve-1* 3'UTR has allowed us to ensure that this is the case. Reviewer #3 has noted that this is a good control in their comments to the authors.

6, The proposed cell-non-autonomous effects of miR-71 in suppressing signaling from muscle to glial cells are intriguing but underexplored. The roles of *ins-11* and *nlp-52* in this process should be validated through RNAi, tissue-specific knockout or knockdown, and tissue-specific rescue assays to understand their function in muscle and UPR^{mt} signaling comprehensively.

We thank the reviewer for their suggestions.

We did already perform RNAi experiments for *ins-11* and *nlp-52* in the submitted manuscript. To build on and strengthen these results, we analysed the induction of glial DVE-1::GFP in deletion mutants of both factors but only observed a significant decrease in the glial signal for *nlp-52(tm12973)* and not for *ins-11(tm1053)* mutants. As such, we decided to focus on NLP-52. To understand the tissue requirements of *nlp-52*, we performed several new experiments.

First, we performed tissue-specific rescue by expressing wild type *nlp-52* under a muscle specific promoter (*myo-3p*) in the *nlp-52(tm12973)* deletion background. Muscle-specific expression of *nlp-52* restored glial DVE-1::GFP signals to control levels (Fig. 6e and f), suggesting that *nlp-52* is sufficient in the BWM to signal muscle-specific mtDSB to the glia.

Second, we performed tissue-specific knockdown of *nlp-52* to determine whether it is required in the muscle to communicate muscle-specific mtDSBs to the glia. We made strains where *nlp-52* was suppressed by RNAi only in the BWM through the expression of *nlp-52* sense and antisense sequences under the *myo-3* promoter. Muscle-specific RNAi of *nlp-52* suppressed DVE-1::GFP activation in the glia (Fig. 6g and h), suggesting that NLP-52 is required in muscle cells to communicate mitochondrial stress localised to the muscle cells to the glia.

Together these new results further strengthen and support our conclusions that miR-71 regulates muscle-to-glia cell-non-autonomous UPR^{mt} activation via the regulation of *nlp-52* in the muscle cells. In light of the new data and lack of support for a role for *ins-11*, we have decided to only include *nlp-52* in the manuscript.

In summary, the study in its current form lacks the significance and depth required to advance our understanding of UPR^{mt} regulation. Addressing these critical points would enhance the mechanistic insight and position of miR-71 within the established landscape of UPR^{mt} regulatory pathways.

We thank the Reviewer for all of their important comments and suggestions. We believe we have now addressed these and as a result enhanced the manuscript by improving the mechanistic understanding of a new miRNA-mediated pathway that regulates the UPR^{mt}.

Reviewer #3 (Remarks to the Author):

The study by Kirmes et al. describes the identification of miR-71 in muscles in response to severe mitochondrial stress and activation of DAF-16, HIF-1 and ATFS-1 and leads to degradation of DVE-1 to alleviate overactivation of the UPR^{mt}. Additionally, to the impact of miR-71 in the muscle, they propose that miR-71 leads to reduced secretion of neuro and insulin-like peptides which act to suppress muscle to glia axis of mitochondrial stress. Therefore, they suggest that miR-71 is a novel target of the UPR^{mt}, that act to prevent its overactivation and leads to local and systemic beneficial effects.

Overall, the introduction is well written and provide appropriate background and relevant literature. The inclusion of a nuclease dead control in the original screen is a strong control, data supporting the impact of miR71 following mitochondrial damage in muscle is strong,

the results linking miR71 to the regulation of DVE1 in figure 2 is convincing and involves proper controls, same applies to all figures notably the control in figure 5 for regulation of UPRmt rather than reporter directly, that miR71 does not spread itself across tissues. Therefore, the study present compelling results and is very well designed.

While the non cell-autonomous effect of mitochondrial stress in one tissue affecting another is already known, the novelty here is to add miR71 and the fact that the strength of signal is important.

We thank Reviewer #3 for their comments and enthusiasm for the manuscript.

Critiques:

1, miR-35 is activated significantly more than miR-71, miR-246 seems activated at similar level to miR-71, several stressors also activates mir35 but polG represses it , which is interested. Is there a specificity within the miR35 family? the focus on miR-71 or lack of analysis of miR-35 and miR-246 needs to be justified in a stronger manner, especially that the group have published in miR-71 previously and therefore the choice of focusing on this particular miR come across as purely biased. The same applies to the focus on the peptides already known to regulate non-cell autonomous communication between tissues following mitochondrial stress in the screen in figure 6.

We thank the reviewer for their comments and questions.

We do not believe there is specificity within the miR-35 family, all members of the miR-35 family in *Caenorhabditis elegans* are transcribed as a single polycistronic primary transcript encoded within an approximately 800-nucleotide cluster, indicating they are regulated by the same promoter. This transcript is processed into single miRNAs (*mir-35-41*) which all share the same seed region¹¹. However, it is an interesting question that may warrant future study.

As noted on page 6, only miR-71 was consistently induced by multiple forms of mtDNA stress, which is why we focused on this particular miRNA. Specifically, miR-71 was induced by the presence of the mtDNA mutation *mpt1* (Fig. 1f) as well as by a mutation in the proof-reading domain of *polg-1* (Fig. 1g). miR-35 was not induced under either of these conditions (Extended Data Fig. 1c and d). Interestingly, miR-71 was also much stronger induced than miR-35 by another mtDNA mutation, *uaDf5* (~9-fold versus ~5-fold; Fig. 1e and Extended Data Fig. 1b). Because the response to mtDNA stress was consistent for miR-71 and not miR-35, we postulated that miR-71 was the most biological significant miRNA in mtDNA damage processes and therefore chose to focus on its mechanism of action.

In the case of miR-246, we did not observe a protective effect against mtDSBs when we overexpressed it and therefore did not focus on its action any further.

We would like to clarify that we have not worked on miR-71 prior to this study. In an unbiased sequencing screen for Argonaute-loaded miRNAs, miR-71 was detected among various other miRNAs to be loaded onto Argonaute proteins isolated from muscle cells (Brosnan et al., 2021)¹², but we did not study its function.

In the case of the peptides, we are unsure what the reviewer is referring to, since to our knowledge, NLP-52 has not previously been reported to play a role in cell-non-autonomous mitochondrial stress signalling. We did note in the text that other, previously characterised peptides involved cell-non-autonomous mitochondrial stress signalling (specifically INS-27, INS-35, FLP-1 and FLP-2) were revealed by our RNAseq experiment being potentially regulated by miR-71 during mitochondrial stress. However, INS-27, INS-35, FLP-1 or FLP-2 were not required for muscle-to-glia mitochondrial stress signalling (Extended Data Table 3).

2, Overexpression of miR-71 increases *daf-2* significantly the receptor of insulin in muscle but this is not discussed, how it relates to the finding of insulin-like peptide in figure 6 (Extended data Fig.2).

We thank the reviewer for raising this interesting point. We do not have an explanation as to why overexpression of miR-71 increases the fluorescence signal of the *daf-2* 3'UTR reporter. Admittedly, this effect is the opposite of anticipated miRNA activities and may therefore be caused indirectly. It is true, that in the light of our findings around the insulin/IGF-1 signalling (IIS) pathway, this observation might gain in importance; our data shows that *mir-71* is transcriptionally regulated by the insulin/IGF-1 signalling (IIS) factor DAF-16 (see Fig. 4), which is downstream of DAF-2. Moreover, mitochondrial stress and miR-71 impacts the levels of several insulin signalling peptides (Extended Data Fig. 5a).

3, Choice of sodium azide for RNAseq is unfortunate and detract from main finding using a mtDNA specific DNA damage. While some validation is provided in fig. 6d and e, a more comparison of the gene expression in panel b and c following muscle and mitochondrial specific DNA damage should be provided.

We thank the reviewer for their suggestion. It is important to note that RNAseq was performed on whole animals, and we hypothesised that systemic mitochondrial stress induced by sodium azide treatment would enable us to detect important downstream targets of miR-71 more easily than tissue-specific mtDSBs, which may be masked by the absence of mitochondrial stress in all other tissues. As the Reviewer suggests, the experiments performed in Figs. 6c and d validate our RNAseq data in the context of muscle-specific mtDSBs and glia cell-non-autonomous signalling. However, to determine if it is possible to detect changes in *ins-11* and *nlp-52* transcripts levels on whole animals experiencing muscle-specific mtDSBs, we performed qPCR. As the results below show we cannot detect an increase in *ins-11* or *nlp-52* in *mir-71* deletion mutants when the mitochondrial stress is restricted to body wall muscle.

Figure 2. Quantitative (q)PCR of (a) *ins-11* and (b) *nlp-52* levels in animals that have mtDSBs induced in muscle cells in a mir-71 deletion background. Columns represent mean \pm SEM; $n \geq 3$; unpaired, two-tailed t-test.

References

- 1 Dai, C. Y. *et al.* ATFS-1 counteracts mitochondrial DNA damage by promoting repair over transcription. *Nature Cell Biology* **25**, 1111-1120 (2023). <https://doi.org/10.1038/s41556-023-01192-y>
- 2 Nargund, A. M., Fiorese, C. J., Pellegrino, M. W., Deng, P. & Haynes, C. M. Mitochondrial and nuclear accumulation of the transcription factor ATFS-1 promotes OXPHOS recovery during the UPR(mt). *Mol Cell* **58**, 123-133 (2015). <https://doi.org/10.1016/j.molcel.2015.02.008>
- 3 Henderson, S. T. & Johnson, T. E. daf-16 integrates developmental and environmental inputs to mediate aging in the nematode *Caenorhabditis elegans*. *Curr Biol* **11**, 1975-1980 (2001). [https://doi.org/10.1016/s0960-9822\(01\)00594-2](https://doi.org/10.1016/s0960-9822(01)00594-2)
- 4 Zhang, Y., Shao, Z., Zhai, Z., Shen, C. & Powell-Coffman, J. A. The HIF-1 hypoxia-inducible factor modulates lifespan in *C. elegans*. *PLoS One* **4**, e6348 (2009). <https://doi.org/10.1371/journal.pone.0006348>
- 5 Rauthan, M., Ranji, P., Aguilera Pradenas, N., Pitot, C. & Pilon, M. The mitochondrial unfolded protein response activator ATFS-1 protects cells from inhibition of the mevalonate pathway. *Proc Natl Acad Sci U S A* **110**, 5981-5986 (2013). <https://doi.org/10.1073/pnas.1218778110>
- 6 Lin, K., Hsin, H., Libina, N. & Kenyon, C. Regulation of the *Caenorhabditis elegans* longevity protein DAF-16 by insulin/IGF-1 and germline signaling. *Nat Genet* **28**, 139-145 (2001). <https://doi.org/10.1038/88850>
- 7 Nargund, A. M., Pellegrino, M. W., Fiorese, C. J., Baker, B. M. & Haynes, C. M. Mitochondrial import efficiency of ATFS-1 regulates mitochondrial UPR activation. *Science* **337**, 587-590 (2012). <https://doi.org/science.1223560> [pii]10.1126/science.1223560
- 8 Senchuk, M. M. *et al.* Activation of DAF-16/FOXO by reactive oxygen species contributes to longevity in long-lived mitochondrial mutants in *Caenorhabditis elegans*. *PLoS Genet* **14**, e1007268 (2018). <https://doi.org/10.1371/journal.pgen.1007268>
- 9 Selak, M. A. *et al.* Succinate links TCA cycle dysfunction to oncogenesis by inhibiting HIF-alpha prolyl hydroxylase. *Cancer Cell* **7**, 77-85 (2005). <https://doi.org/10.1016/j.ccr.2004.11.022>
- 10 Lee, S. J., Hwang, A. B. & Kenyon, C. Inhibition of respiration extends *C. elegans* life span via reactive oxygen species that increase HIF-1 activity. *Curr Biol* **20**, 2131-2136 (2010). <https://doi.org/10.1016/j.cub.2010.10.057>
- 11 Massirer, K. B., Perez, S. G., Mondol, V. & Pasquinelli, A. E. The miR-35-41 family of microRNAs regulates RNAi sensitivity in *Caenorhabditis elegans*. *PLoS Genet* **8**, e1002536 (2012). <https://doi.org/10.1371/journal.pgen.1002536>

- 12 Brosnan, C. A., Palmer, A. J. & Zuryn, S. Cell-type-specific profiling of loaded miRNAs from reveals spatial and temporal flexibility in Argonaute loading. *Nature Communications* **12** (2021). <https://doi.org/ARTN 219410.1038/s41467-021-22503-7>